# Autocatalytic activation of a malarial egress protease is druggable and requires a protein cofactor

Michele S Y Tan[1] , Konstantinos Koussis[1], Chrislaine Withers-Martinez[1], Steven A Howell[2], James A Thomas[3] , Fiona Hackett[1], Ellen Knuepfer[4], Min Shen[5], Matthew D Hall[5], Ambrosius P Snijders[2] & Michael J Blackman[1,3,*]

## Abstract

Malaria parasite egress from host erythrocytes (RBCs) is regulated by discharge of a parasite serine protease called SUB1 into the parasitophorous vacuole (PV). There, SUB1 activates a PV-resident cysteine protease called SERA6, enabling host RBC rupture through SERA6-mediated degradation of the RBC cytoskeleton protein β-spectrin. Here, we show that the activation of *Plasmodium falciparum* SERA6 involves a second, autocatalytic step that is triggered by SUB1 cleavage. Unexpectedly, autoproteolytic maturation of SERA6 requires interaction in multimolecular complexes with a distinct PV-located protein cofactor, MSA180, that is itself a SUB1 substrate. Genetic ablation of MSA180 mimics SERA6 disruption, producing a fatal block in β-spectrin cleavage and RBC rupture. Drug-like inhibitors of SERA6 autoprocessing similarly prevent β-spectrin cleavage and egress in both *P. falciparum* and the emerging zoonotic pathogen *P. knowlesi*. Our results elucidate the egress pathway and identify SERA6 as a target for a new class of antimalarial drugs designed to prevent disease progression.

**Keywords** cofactor; egress; malaria; *Plasmodium falciparum*; protease
**Subject Categories** Microbiology, Virology & Host Pathogen Interaction; Post-translational Modifications & Proteolysis
**The EMBO Journal (2021) 40: e107226**

## Introduction

Malaria is a devastating infectious disease caused by protozoan parasites of the genus *Plasmodium*. The absence of an efficacious vaccine, compounded by widespread resistance to frontline chemotherapies, necessitates an improved understanding of malaria parasite biology to identify new drug targets. Clinical disease arises from replication of the malaria parasite within an intraerythrocytic membrane-bound parasitophorous vacuole (PV). In *P. falciparum*, the deadliest parasite species, each replication cycle lasts ~48 h, culminating in the formation of a multinucleated schizont that undergoes segmentation to generate daughter merozoites. These are released from the host cell in a lytic process called egress to invade fresh RBCs.

Merozoite egress is a tightly regulated, protease-dependent process comprising several rapidly successive steps to achieve sequential rupture of the PV membrane (PVM) and host RBC membrane (RBCM). Minutes before egress, the activation of the single parasite cyclic GMP-dependent protein kinase (PKG) triggers the discharge of a subtilisin-like protease called SUB1 from merozoite secretory organelles called exonemes into the PV lumen (Collins *et al*, 2013b), where SUB1 proteolytically processes several PV and merozoite surface proteins (Yeoh *et al*, 2007; Koussis *et al*, 2009; De Monerri *et al*, 2011). SUB1 is essential for all the morphological changes associated with egress, including PVM rupture and RBCM permeabilization (poration), rapidly followed by RBCM rupture (Thomas *et al*, 2018). Among known *P. falciparum* SUB1 substrates is serine repeat antigen 6 (SERA6), a cysteine protease-like protein with orthologues in all *Plasmodium* species. SERA6 is essential specifically for the final step of egress, RBCM rupture, where it acts by mediating proteolytic cleavage of the major RBC cytoskeletal protein β-spectrin, leading to destabilisation of the host cell cytoskeleton and loss of structural integrity of the RBCM (Thomas *et al*, 2018).

SERA6 is a member of a multigene family encoding proteins all of which possess a central domain with homology to papain-like (clan CA, family C1) cysteine peptidases (Arisue *et al*, 2011). Of the nine *P. falciparum* SERA orthologues, only SERA5 and SERA6 have been implicated in asexual blood stage egress. Whilst SERA6 is essential (Thomas *et al*, 2018), SERA5 plays an important but

1   Malaria Biochemistry Laboratory, The Francis Crick Institute, London, UK
2   Protein Analysis and Proteomics Platform, The Francis Crick Institute, London, UK
3   Faculty of Infectious and Tropical Diseases, London School of Hygiene & Tropical Medicine, London, UK
4   Department of Pathobiology and Population Sciences, Royal Veterinary College, Hertfordshire, UK
5   National Center for Advancing Translational Sciences (NCATS), National Institutes of Health, Rockville, MD, USA
    *Corresponding author. Tel: +44 0203 796 2328; E-mail: mike.blackman@crick.ac.uk

non-essential (Collins *et al*, 2017) and non-enzymatic (Stallmach *et al*, 2015) role. SERA5 is processed by SUB1 at two discrete sites flanking its central papain-like domain. Sequence alignments between the two SERAs and experimental cleavage of short synthetic peptides (Yeoh *et al*, 2007) as well as recombinant SERA6 (Ruecker *et al*, 2012) collectively suggest that SUB1 processing of SERA6 similarly releases its central papain-like domain and so this has been postulated to represent a proteolytic activation event. Neither a catalytically inactive SERA6 mutant (in which the predicted catalytic Cys644 residue was substituted with Ala) nor a SERA6 mutant possessing mutations at the two SUB1 processing sites that rendered them uncleavable (Ruecker *et al*, 2012) were capable of rescuing a fatal SERA6-null (*ΔSERA6*) egress defect (Thomas *et al*, 2018). Collectively, these findings all indicate that SERA6 is a cysteine protease that plays an essential catalytic role in egress that is regulated by SUB1 processing. However, whether SERA6 maturation might involve steps beyond SUB1 processing is unknown.

Here, we demonstrate that SUB1 cleavage of SERA6 is in fact a trigger for a second, autoprocessing step required to complete SERA6 maturation. In addition, we report the discovery of a protein cofactor of SERA6 that is essential for its proteolytic activity and its function in mediating RBCM rupture. Finally, we identify potent drug-like inhibitors of egress that act through direct inhibition of SERA6 autocatalytic maturation.

# Results

## Epitope tagging of SERA6 enables monitoring of SERA6 expression and localisation

We have previously examined SERA6 processing in mature *P. falciparum* schizonts (Ruecker *et al*, 2012), but attempts to fully characterise the processed forms of SERA6 generated during egress were impeded by lack of specific antibodies and the inability to fuse an N- or C-terminal epitope tag to SERA6. SERA5 is similarly resistant to C-terminal fusion of an epitope tag, but does tolerate an internal tag (Stallmach *et al*, 2015). Reasoning that a similar approach might be applicable to SERA6, we used Cas9-enhanced marker-free homologous recombination to modify the *SERA6* locus in the DiCre-expressing *P. falciparum* B11 line (Perrin *et al*, 2018) to simultaneously achieve: (i) insertion of an internal "mini" tandem affinity-purification (mTAP) tag consisting of a triple hemagglutinin (HA3) epitope tag and a Strep-tag II sequence, followed by a tobacco etch virus (TEV) protease cleavage motif (Stallmach *et al*, 2015) just downstream of the central papain-like domain (Fig 1A, top) and (ii) insertion of a *loxP*-containing heterologous intron (*loxPint*) (Jones *et al*, 2016) into the papain-like domain (Fig EV1A). The latter step took advantage of the single *loxP* site immediately downstream of the upstream *SERA5* gene already present in the parental B11 genome, effectively flanking with *loxP* sites a segment of *SERA6* including the catalytic Cys644 codon.

Clonal lines of the modified parasites (called *SERA6-mTAP:loxP*) were generated by limiting dilution and the structure of the modified *SERA6* locus verified by diagnostic PCR (Fig EV1B). Examination of mature *SERA6-mTAP:loxP* schizonts by Western blot and immunofluorescence assay (IFA) confirmed expression and localisation of the

tagged protein to the PV lumen as expected (Fig EV1C and D). Treatment of *SERA6-mTAP:loxP* parasites with rapamycin (RAP) to induce DiCre activity resulted in the expected excision of the floxed DNA sequence (Fig EV1B) and loss of SERA6-mTAP expression (Fig EV1C and D) by the end of the erythrocytic cycle in which the parasites were RAP-treated (cycle 0), with the expected failure to proliferate (Fig EV1E) and blockade of egress (Fig EV1F and G). Importantly, mock-treated *SERA6-mTAP:loxP* parasites showed normal growth rates relative to the parental B11 line (Fig EV1E), confirming correct functioning of the epitope-tagged SERA6.

## SERA6 is rapidly proteolytically processed into multiple protein species at egress

To characterise the maturation of SERA6, we examined the fate of SERA6-mTAP upon egress. For this, *SERA6-mTAP:loxP* schizonts were incubated for up to 4 h with the reversible PKG inhibitor (4-[7-[(dimethylamino)methyl]-2-(4-fluorphenyl)imidazo[1,2-α]pyridine-3-yl] pyrimidin-2-amine (compound 2; C2) which prevents egress, arresting the parasites in a state of full maturation. Washout of this compound rapidly relieves PKG inhibition, allowing SUB1 discharge and egress of the majority of the parasite population to proceed within ~10–20 min (Collins *et al*, 2013b; Thomas *et al*, 2018). The use of C2 in this way thus provides an exquisite degree of control over the timing of egress. Western blot analysis showed that in mature, C2-arrested *SERA6-mTAP:loxP* schizonts, SERA6-mTAP was present as a presumably full-length ~125 kDa protein (Fig 1A, 0 min). Upon washout of C2, allowing SUB1 discharge into the PV lumen, SERA6-mTAP was rapidly converted to a ~70 kDa form which retained the internal epitope tag, and subsequently into several smaller forms (Fig 1A). The 70 kDa species closely matches the predicted molecular mass (65 kDa) of the central segment of SERA6 generated by SUB1-mediated proteolysis at the two known cleavage sites (Ruecker *et al*, 2012) with addition of the ~6 kDa mTAP tag. This processed species is therefore referred to as p65-mTAP. The smaller SERA6 species produced comprised ~50, ~40 and ~36 kDa forms which also retained the internal tag (referred to as Intermediate-mTAP, Intermediate(II)-mTAP and PAP-mTAP, respectively) as well as a distinct ~40 kDa product (called p40) that does not contain the tag and was only detected by a polyclonal anti-SERA6 antibody called S6C1 (Ruecker *et al*, 2012) (Fig 1A). The predicted mass of the tagged papain-like domain is ~44 kDa, so it was considered likely that PAP-mTAP, the major terminal processed species, effectively comprises the papain-like domain. Based on the known binding properties of the S6C1 antibody (Ruecker *et al*, 2012), the p40 product most likely spans the region between the SUB1 site 1 (Fig 1A) and the papain-like domain.

To more precisely determine the composition of the various SERA6-mTAP species produced at egress, the mTAP tag was exploited to perform immunoprecipitation from extracts of mature schizonts sampled at intervals over the course of egress, followed by mass spectrometric analysis (Fig 1B). This confirmed that p65-mTAP results from SUB1-mediated cleavage of full-length SERA6 at sites 1 and 2. The analysis further confirmed that PAP-mTAP contained most of the papain-like domain, crucially including all three catalytic triad residues (Fig 1C, fragment H); PAP-mTAP thus likely represents the terminal activated form of SERA6. The C terminus of PAP-mTAP included the HA3 epitope, but lacked the adjacent

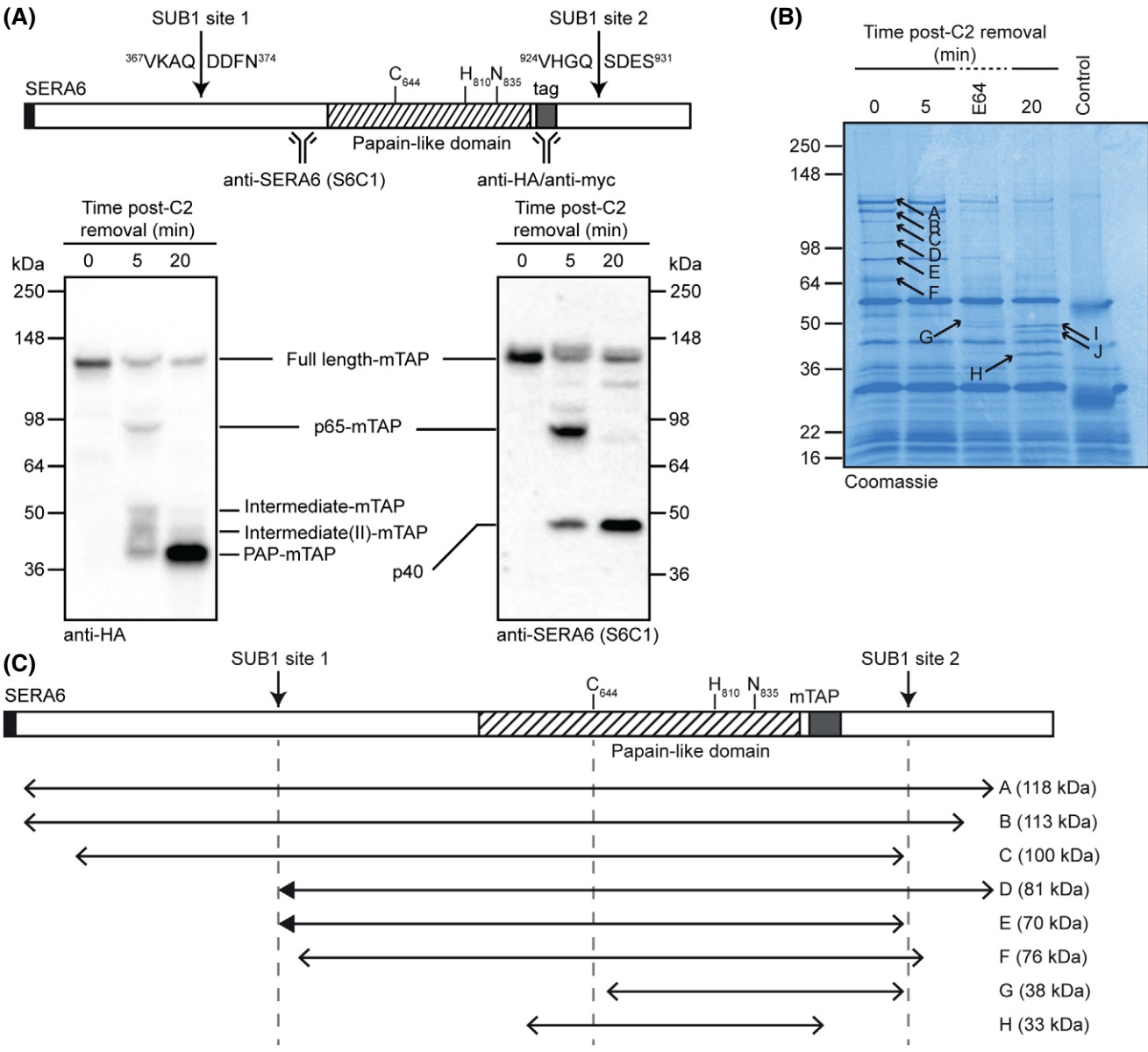

**Figure 1. SERA6 is rapidly proteolytically processed into multiple protein species at egress.**

A   Top: SERA6 architecture. Positions of the papain-like domain, catalytic triad residues, SUB1 cleavage sites and epitopes for the antibodies used are indicated. Positions of the inserted mTAP or myc3 epitope tags (between codons Asn[886] and Val[887]) introduced in *SERA6-mTAP:loxP* parasites are shown. Also see Fig EV1. Bottom: time-course analysis of egress by Western blot, showing processing of SERA6-mTAP into multiple protein species (representative of 2 independent experiments). Schizonts were sampled at the indicated times following washout of C2-mediated arrest. Nomenclature for each protein species was based on apparent molecular mass or predicted composition. Note that the predicted mass of the p40 species is ~27 kDa, but its migration on SDS–PAGE may be aberrant due to its acidic nature; the predicted pI of sequence between SUB1 site 1 and the central papain-like domain (Asp371-Lys605) is ~4.4 (see https://web.expasy.org/compute_pi/).

B   Coomassie-stained SDS–PAGE showing proteins immunoprecipitated from schizonts expressing SERA6-mTAP, sampled at the indicated times following C2 washout or arrested with E64 (50 μM). Control, proteins immunoprecipitated from *SERA6:loxP* (Thomas *et al*, 2018) schizonts, which express non-epitope-tagged SERA6, sampled at 5 min following C2 washout. Arrowed, species analysed by mass spectrometric peptide mapping. Fragments A-H were identified as SERA6-mTAP-derived, whilst fragments I and J were derived from a different protein, PF3D7_1014100.

C   Schematic of the likely identities of the different SERA6-mTAP species identified from peptide mapping, relative to full-length SERA6-mTAP and the papain-like domain (hatched). The predicted mass of each species based on its amino acid composition is indicated. Detection of a semi-tryptic SERA6-derived peptide (filled block arrowhead) unambiguously confirmed cleavage at SUB1 site 1 whilst the coverage allowed mapping of up to five residues upstream of SUB1 site 2 (fragment E). No evidence was found for processing at a third putative SUB1 cleavage site (Ruecker *et al*, 2012). The N-terminus of fragment H (assumed to correspond to PAP-mTAP) could not be precisely mapped; attempts to assign its N-terminus by Edman degradation were unsuccessful. Also see Fig EV2.

Source data are available online for this figure.

Strep-tag II sequence and TEV protease cleavage motif (Fig EV2), indicating "trimming" of the mTAP tag. Collectively, these results indicated that conversion of full-length SERA6 to the p65 form results directly from cleavage by SUB1 at sites 1 and 2. However, the data also showed that SERA6 maturation involves further cleavage events that are not mediated by SUB1, implicating one or more additional protease activities.

**The SERA6 p65-to-PAP maturation step is autocatalytic**

Many cysteine proteases undergo autocatalytic cleavage *in trans* or *in cis* (intramolecular) during activation (Turk *et al*, 2012; Verma *et al*, 2016). To explore whether SERA6 activity plays a role in its maturation, we designed plasmids for episomal expression of wild-type (WT; control) or catalytically dead (Cys644Ala) SERA6 (Fig 2A) and introduced these transgenes into *SERA6-mTAP:loxP* parasites. To detect their expression on the background of the endogenous SERA6-mTAP, both transgenes included an inserted triple myc sequence (myc3) in place of the mTAP tag. The resulting parasite lines, referred to as *SERA6-WT-myc3 [SERA6-mTAP:loxP]* and *SERA6-Cys644Ala-myc3 [SERA6-mTAP:loxP]*, expressed both their respective episome-encoded myc3-tagged SERA6 transgene products as well as the endogenous SERA6-mTAP (Fig 2B, 0 min). Upon RAP treatment of the parasites, the expression of the chromosomal SERA6-mTAP was selectively silenced, as expected (Fig 2B, 0 min), enabling us to monitor the fate and function of the episomal transgene products.

In the presence of endogenous SERA6-mTAP, both the SERA6-WT-myc3 and SERA6-Cys644Ala-myc3 mutants were converted to the p65-myc3 species and two smaller tagged forms of ~50 and ~40 kDa (referred to as Intermediate-myc3 and PAP-myc3, respectively) (Fig 2B, DMSO lanes). No ~36 kDa form of the myc3-tagged mutants was detected, despite processing of SERA6-mTAP into the terminal ~36 kDa PAP-mTAP species being unaffected. This and the previously observed "trimming" of the mTAP tag in SERA6 PAP-mTAP (Fig EV2) suggested that the mTAP tag undergoes artefactual cleavage following maturation of SERA6-mTAP and that the ~40 kDa tagged species represents the correctly processed form of SERA6. Nevertheless, as parasites expressing exclusively SERA6-mTAP are viable (Fig EV1E), this trimming step is clearly not detrimental to the function of the protein. Production of the p40 species was not affected by the type of tag incorporated (Fig 2B, anti-SERA6 blots).

To examine maturation of the myc3-tagged SERA6 transgene products in the absence of endogenous SERA6-mTAP, we RAP-treated the parasite lines to abolish SERA6-mTAP expression then monitored the fate of the ectopically expressed myc3-tagged proteins. Processing of SERA6-WT-myc3 to the terminal PAP-myc3 form was unaffected by RAP treatment of the *SERA6-WT-myc3 [SERA6-mTAP:loxP]* parasites (Fig 2B, SERA6-WT-myc3 RAP lanes). In contrast, in RAP-treated *SERA6-Cys644Ala-myc3 [SERA6-mTAP:loxP]* parasites, whilst processing of SERA6-Cys644Ala-myc3 to p65-myc3 was unaffected, further conversion of p65-myc3 to the Intermediate-myc3 and p40 appeared less efficient and no PAP-myc3 was formed (Fig 2B, SERA6-Cys644Ala-myc3 RAP lanes). This maturation defect was mimicked by treating schizonts with the broad-spectrum membrane-permeable cysteine protease inhibitor (2S,3S)-trans-epoxysuccinyl-L-leucylamido-3-methylbutane ethyl ester (E64-d), a potent inhibitor of egress which mimics SERA6 disruption by blocking RBCM rupture whilst allowing PVM rupture (Chandramohanadas *et al*, 2011) (Fig 2B, E64-d lanes). These findings strongly suggested that conversion of p65 to the terminal PAP form requires enzymatically active SERA6, indicating an autocatalytic mechanism. Additionally, the efficient maturation of SERA6-Cys644Ala-myc3 to PAP-myc3 in the presence of SERA6-mTAP but incomplete maturation observed in the absence of a functional chromosomal gene suggests that this autoprocessing can occur *in trans*. Importantly, ablation of SERA6 catalytic function or treatment with E64-d had no effect under any conditions on conversion of full-length SERA6 to the p65 species, consistent with this step being mediated by SUB1 (which is not sensitive to E64-d). Conversion to the intermediate forms also occurred under these conditions; this too may be a consequence of SUB1-mediated cleavage at additional sites.

Importantly, following RAP treatment of the episome-transfected parasite lines, egress only occurred in the case of *SERA6-WT-myc3 [SERA6-mTAP:loxP]*, where a functional form of SERA6 was still expressed and did not occur in the RAP-treated *SERA6-Cys644Ala-myc3 [SERA6-mTAP:loxP]* parasites (Fig 2C). In concordance with this and the profiles of SERA6 maturation in *SERA6-WT-myc3 [SERA6-mTAP:loxP]* and *SERA6-Cys644Ala-myc3 [SERA6-mTAP:loxP]* parasites, host RBC β-spectrin cleavage was only observed when at least one functional variant of SERA6 was expressed, but was notably absent when only the catalytically inactive SERA6 Cys644Ala mutant was expressed (Fig 2C). We concluded that conversion of SERA6 to the terminal PAP form is crucial for its function in dismantling the host RBC cytoskeleton at egress. Collectively, these findings demonstrate that maturation of SERA6 is a multi-step process which is initiated by SUB1 but which involves a second,

---

**Figure 2.  Maturation of SERA6 p65 to the terminal PAP form is autocatalytic.**

A  Complementation constructs introduced into *SERA6-mTAP:loxP* parasites for episomal expression of wild-type (WT) or catalytically dead (Cys644Ala) SERA6. The papain-like domain (hatched) and myc3 epitope tag are indicated. SERA6 expression from these constructs was under the control of its native promoter.

B  Western blot analysis of egress time-course, comparing maturation of the different SERA6 variants from DMSO- and RAP-treated *SERA6-WT-myc3 [SERA6-mTAP:loxP]* and *SERA6-Cys644Ala-myc3 [SERA6-mTAP:loxP]* parasites (representative of 2 independent experiments). Schizonts were sampled at the indicated times following washout of C2, or arrested with E64-d (50 μM), a membrane-permeable derivative of E64 which is a more potent inhibitor of SERA6 maturation (Appendix Fig S1). Samples were probed with anti-myc to detect the myc3-tag, anti-HA to detect the mTAP tag or anti-SERA6 (S6C1) to detect a different region of SERA6, respectively (see Fig 1A for antibody epitope positions). One of the blots was stripped and reprobed with anti-AMA1 as a loading control. Nomenclature for each protein species was based on apparent molecular mass or predicted composition. Also see Fig EV2.

C  Giemsa-stained thin films and Western blot of DMSO- and RAP-treated *SERA6-WT-myc3 [SERA6-mTAP:loxP]* and *SERA6-Cys644Ala-myc3 [SERA6-mTAP:loxP]* parasites showing that cleavage of host RBC β-spectrin (red arrowhead) does not occur unless a functional copy of SERA6 is present, and is tightly associated with egress (reproducible in 2 independent experiments). Schizonts were sampled at the indicated times following washout of C2. Scale bar, 5 μm.

Source data are available online for this figure.

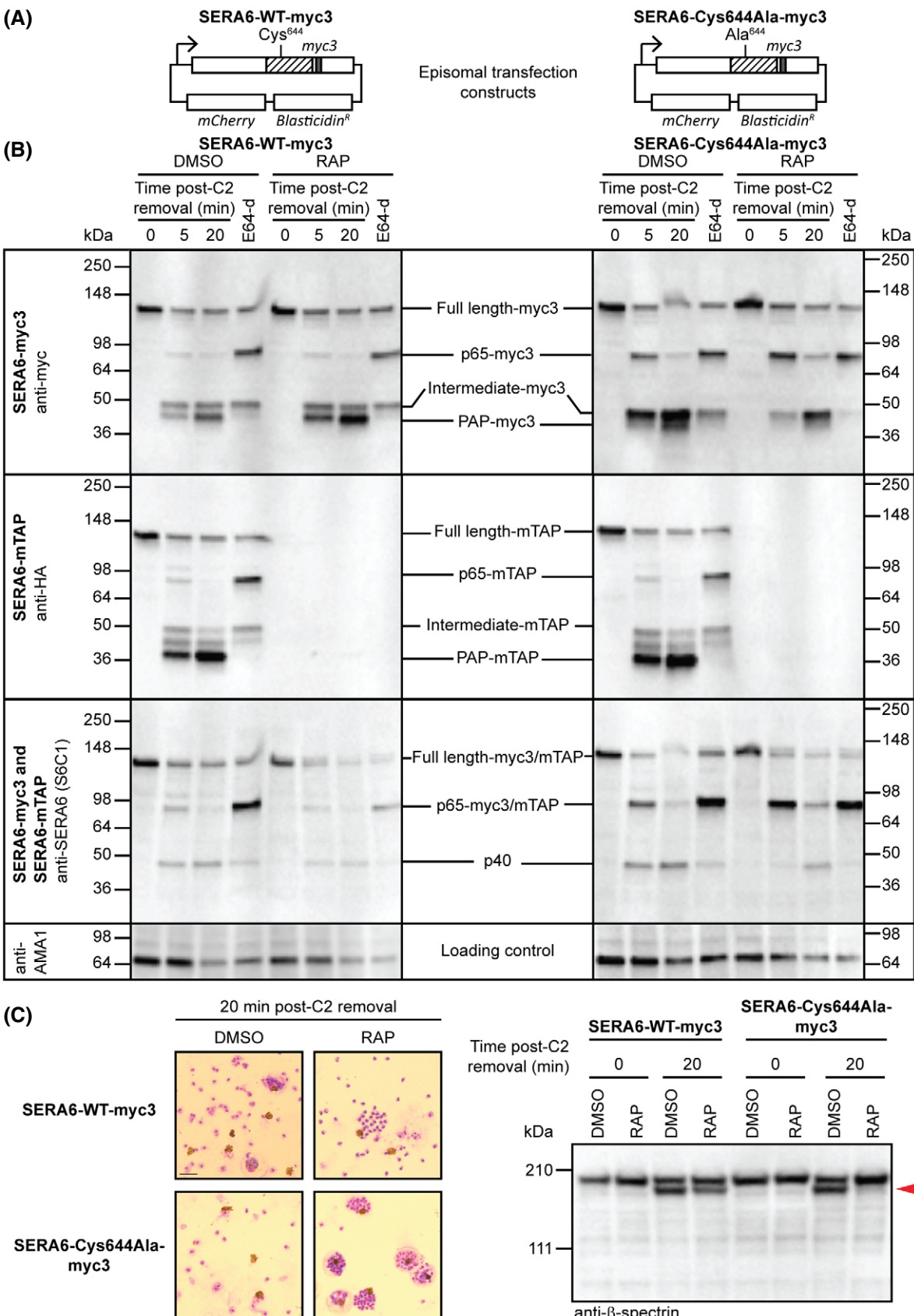

**Figure 2.**

autocatalytic phase that generates the terminal, proteolytically active PAP form required for host RBC β-spectrin cleavage and egress.

## MSA180 is an interaction partner of activated SERA6

In the course of our mass spectrometric analysis of the various SERA6-mTAP species isolated by immunoprecipitation (Fig 1B), we additionally identified two co-precipitating non-contiguous polypeptide fragments of a distinct protein, annotated in PlasmoDB as merozoite surface antigen 180 (MSA180; PF3D7_1014100) (Figs 1B, 3A, and EV3A and B). Remarkably, the two MSA180 fragments (referred to as MSA180 p45 and MSA180 p40, respectively) were only precipitated from extracts of parasites in which SERA6-mTAP had been processed to PAP-mTAP. Furthermore, based on their intensity on Coomassie-stained SDS–PAGE gels (Figs 1B and EV3B), both MSA180-derived proteins were reproducibly pulled-down at approximately similar stoichiometric amounts as PAP-mTAP, collectively suggestive of a specific interaction with the mature form of SERA6.

MSA180 has orthologues in all known *Plasmodium* species. Alignment of predicted primary sequences of orthologues from other human-infecting *Plasmodium* species highlighted three distinct conserved segments (Fig 3A, Appendix Fig S2). MSA180 p40 and MSA180 p45 overlap with the central and C-terminal conserved regions, respectively, suggesting that these putative SERA6-interaction regions may possess conserved function(s). The precise boundaries of MSA180 p40 could not be unambiguously identified from our mass spectrometric data. However, a semi-tryptic peptide likely corresponding to the N-terminus of MSA180 p45 was detected and the coverage of this species was mapped up to two residues from the C-terminal end of the protein (Fig 3A). MSA180 p45 therefore probably extends from Ser1079-Pro1455 (the C-terminus of MSA180).

Previous work by others on MSA180 found that the protein is proteolytically converted to a 45 kDa species, postulated to be mediated by SUB1 (Nagaoka *et al*, 2019). The residues flanking the N-terminus of MSA180 p45 ($^{1075}$VTGD↓SVEN$^{1082}$) indeed bear hallmarks of the consensus *P. falciparum* SUB1 recognition motif, previously determined as Ile/Leu/Val/Thr-Xaa-Gly/Ala-Paa (not Leu) ↓ Xaa (where Xaa is any amino acid residue and Paa tends to be a polar residue) (De Monerri *et al*, 2011; Withers-Martinez *et al*, 2012). To address whether this sequence could be a genuine SUB1 cleavage site, we tested whether recombinant *P. falciparum* SUB1 (rPfSUB1) could correctly cleave a synthetic peptide (Ac-KVTGDSVENI-COOH) based on the sequence spanning the putative site. This showed that the peptide was efficiently cleaved at the Asp-Ser bond upon incubation with rPfSUB1 (Fig EV4), suggesting that MSA180 is indeed an authentic novel SUB1 substrate. Together, these findings suggested that MSA180 undergoes processing by SUB1 to form the MSA180 p40 and p45 species, which then form a specific molecular complex with mature SERA6.

## MSA180 is an essential PV-located cofactor required for maturation of SERA6

Despite the absence of predicted transmembrane or glycolipid anchoring motifs, MSA180 is annotated as a merozoite surface protein and has previously been suggested to play a role in RBC invasion (Muh *et al*, 2017; Nagaoka *et al*, 2019). However, our discovery of its interaction with SERA6 implied a potential role in egress. To dissect the function of MSA180, we explored its subcellular localisation and the consequences of its disruption. For this, we modified the *MSA180* locus in the DiCre-expressing *P. falciparum* B11 line to simultaneously epitope tag the gene and introduce *loxP* sites to enable conditional truncation in a manner designed to

---

**Figure 3. Efficient conditional disruption of MSA180.**

A MSA180 architecture. Predicted signal peptide (black), conserved regions (hatched; see Appendix Fig S2 for sequence alignments), likely boundaries of the SERA6-interacting regions (double-headed arrows; fragments I and J from Fig 1B; See also Fig EV3) and predicted molecular mass of each species based on its amino acid composition (in parentheses) are indicated. A semi-tryptic MSA180-derived peptide (filled block arrowhead) was detected from fragment I and is likely a result of SUB1 cleavage (see Fig EV4).

B Cas9-based approach for conditional disruption of the *MSA180* gene. Sequential modification by marker-free gene editing was used to achieve: (I) insertion of a C-terminal HA3 epitope tag and a *loxP* site just downstream of the stop codon; followed by (II) insertion of a *loxPint* module just upstream of the central conserved region. Modification I produced a *P. falciparum* clone called *MSA180-HA3 [B11]*, which was subsequently modified (II) to generate the *MSA180-HA3:loxP* parasite line. Conserved regions (black bars), sgRNA targeting sites (break), *loxP* sites (arrowheads), recodonised segments of *MSA180* (hatched) and corresponding homology arms (regions linked by grey dotted lines) are shown. Half-arrows, primers used for diagnostic PCR to confirm gene editing and excision of floxed sequences (see Table EV1 for primer sequences).

C Diagnostic PCR confirming modification of the *MSA180* locus in two independent *MSA180-HA3:loxP P. falciparum* clones.

D Diagnostic PCR confirming efficient DiCre-mediated disruption of *MSA180* within the cycle of RAP treatment (cycle 0) (representative of 6 independent experiments).

E Western blots of DMSO- and RAP-treated *MSA180-HA3:loxP* parasites showing successful epitope tagging and RAP-inducible ablation of MSA180-HA3 expression (red arrowhead; reproducible in 2 independent experiments). Mature C2-arrested cycle 0 schizonts were subjected to SDS–PAGE and blots probed with anti-HA and anti-MSA180, respectively. The anti-HA blot was reprobed with anti-AMA1 as a loading control. The numerous species detected by anti-MSA180 in RAP-treated *MSA180-HA3:loxP* extracts are likely non-MSA180-derived proteins cross-reactive with the polyclonal serum used.

F Representative IFA images confirming loss of MSA180-HA3 expression following RAP treatment (reproducible in 2 independent experiments). Mature C2-arrested cycle 0 schizonts were co-stained with anti-HA (red) and anti-SERA5 (green). Merged signals include that of the DNA dye 4,6-diamidino-2-phenylindole (DAPI; blue). Scale bar, 5 μm.

G Western blots showing MSA180 is a predominantly soluble PV protein (representative of 2 independent experiments). Mature C2-arrested *MSA180-HA3:loxP* schizonts were sampled prior to (Total) and following saponin fractionation to separate RBC cytosol plus soluble PV proteins (Supernatant) from other parasite proteins and membrane-associated proteins (Pellet). The same samples were probed with anti-SERA5 and anti-MSP1 (mAb 89.1) as positive controls for soluble PV proteins and merozoite surface parasite proteins, respectively. The presence of MSA180 p45-HA3 is likely a post-lysis artefact, perhaps due to SUB1 release during sample manipulation. Note the less efficient saponin-mediated release of MSA180-HA3 compared with SERA5, indicating limited membrane association of a fraction of MSA180.

Source data are available online for this figure.

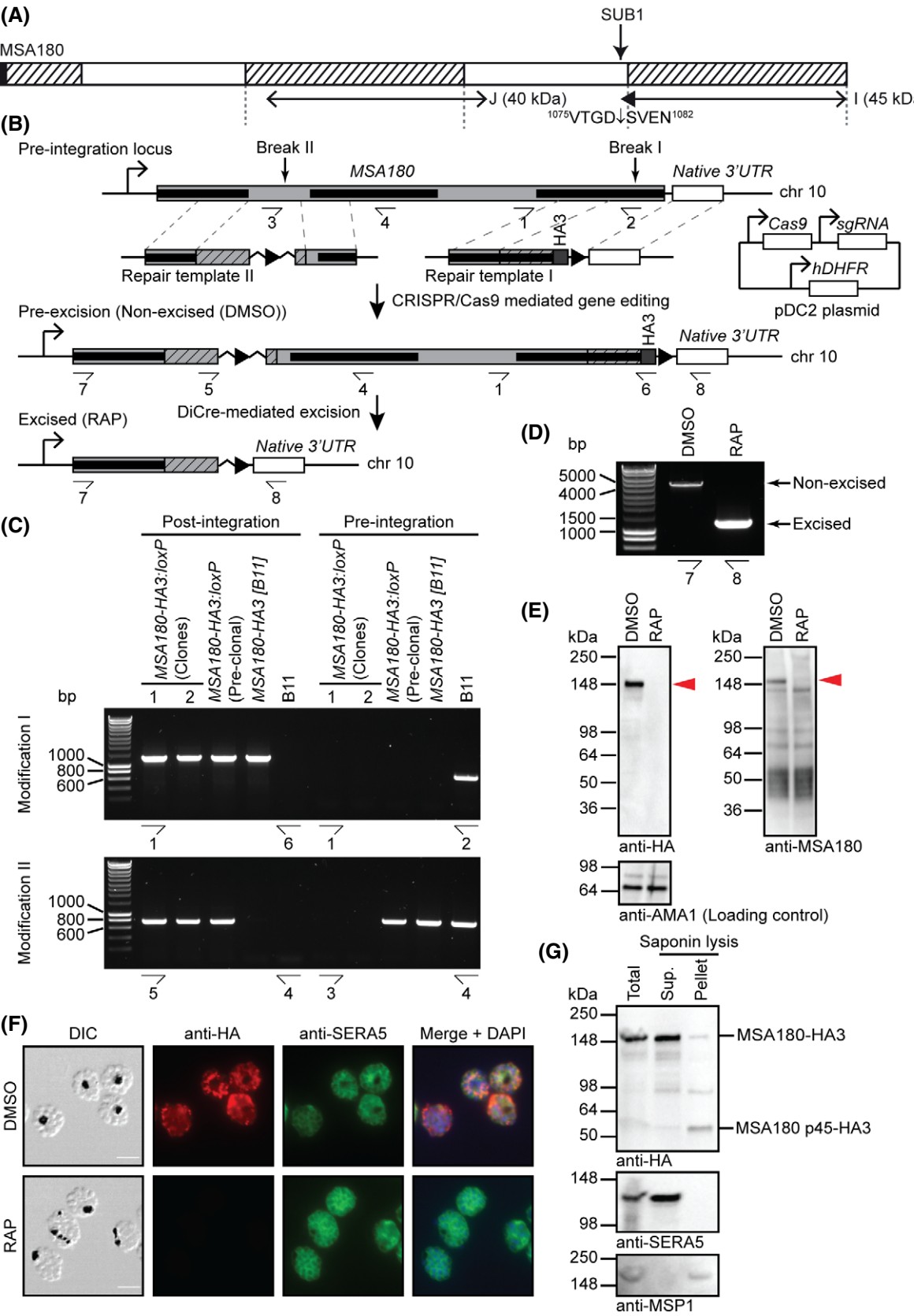

**Figure 3.**

remove the SERA6-interacting segments of the protein (Fig 3B and C). RAP treatment of the modified parasites (called *MSA180-HA3: loxP*) resulted in the expected efficient excision of the floxed gene sequence (Fig 3D). Western blot analysis of mature *MSA180-HA3: loxP* schizonts confirmed the expression of the tagged gene as a ~180 kDa, presumably full-length protein (Fig 3E); this expression was silenced within a single erythrocytic cycle upon RAP treatment (Fig 3E). Examination by IFA of mature *MSA180-HA3:loxP* schizonts likewise confirmed expression of the tagged protein, which disappeared upon treatment with RAP (Fig 3F). Partial co-localisation of the IFA signal with SERA5, a soluble PV protein, suggested localisation of MSA180 to the PV (Fig 3F), in contrast to the previously reported merozoite surface localisation (Nagaoka *et al*, 2019). To further examine this, mature *MSA180-HA3:loxP* schizonts were treated with saponin to selectively rupture both the RBCM and the PVM, and the released soluble and membrane-associated protein constituents examined by Western blot. This showed that, consistent with the IFA signal, MSA180-HA3 was predominantly detected in the soluble saponin-released protein fraction (Fig 3G), suggesting that MSA180 is indeed mostly expressed as a soluble PV protein. However, limited association with the merozoite surface or PVM cannot be ruled out since a small fraction of the MSA180 protein population was consistently detected in the membrane-associated protein fraction.

To determine the essentiality of MSA180, replication of *MSA180-HA3:loxP* parasites was monitored. Whilst mock-treated parasites showed normal growth rates relative to the parental B11 line, RAP-treated parasites failed to proliferate (Fig 4A), indicating that *ΔMSA180* parasites are not viable. In the light of our evidence that MSA180 interacts with mature SERA6 in the PV, we hypothesised that the inability of *ΔMSA180* parasites to replicate was likely due to an egress defect similar to that characteristic of *ΔSERA6* parasites. Upon PVM rupture but just prior to RBCM rupture at egress, the RBCM abruptly becomes permeable or porated, allowing passage of small molecules including the F-actin-binding peptide phalloidin (Glushakova *et al*, 2010; Collins *et al*, 2017). In previous work, the RBCM of *ΔSERA6* parasites has been shown to undergo poration despite the lack of RBCM rupture and merozoite release (Thomas *et al*, 2018). To examine whether *ΔMSA180* parasites displayed a similar phenotype, we used simultaneous differential interference contrast (DIC) and fluorescence live time-lapse microscopy to monitor mature RAP-treated *MSA180-HA3:loxP* parasites as they approached egress at the end of cycle 0. This revealed a defect in host RBCM rupture, despite apparently normal rounding up of the bounding RBCM, PVM rupture and RBCM poration (Fig 4B). This egress phenotype is indistinguishable from that displayed by *ΔSERA6* schizonts. Consistent with these observed phenotypic similarities between *ΔMSA180* and *ΔSERA6* parasites, RAP-treated *MSA180-HA3:loxP* parasites also showed a defect in processing of host β-spectrin (Fig 4C).

These results suggested that the role of MSA180 at egress is related to the function of its SERA6 partner protein. To further explore this, we examined maturation of SERA6 in *ΔMSA180* parasites. As shown in Fig 4D, conversion of the full-length SERA6 precursor to p65 was unaffected, indicating no impact of MSA180 disruption on SUB1 activity. However, further maturation of SERA6 p65 was severely impaired as shown by extended accumulation of the p65 species. The SERA6 maturation defect in *ΔMSA180*

schizonts was remarkably similar to that resulting from treatment with E64-d, or in schizonts expressing a catalytically dead SERA6 mutant, where the autocatalytic maturation steps downstream of p65 generation are defective (Fig 4D). We concluded that MSA180 is an essential, PV-located protein cofactor that facilitates the auto-proteolytic maturation of SERA6 (Fig 4E). We propose renaming MSA180 to Malarial SERA Activator 180 to reflect this newly identified function.

## SERA6 and its partner MSA180 form transient multimolecular condensates during egress

Incorporation of the internal mTAP tag in *SERA6-mTAP:loxP* parasites provided an avenue for tracking the subcellular localisation of SERA6 during the dynamic morphological transitions that occur at egress. To explore this, *SERA6-mTAP:loxP* schizonts sampled at different time points during egress were examined by IFA. Prior to SUB1 discharge, a strong diffuse HA signal was observed throughout the schizont, presumably localised to the PV as it did not completely fill the infected RBC (Fig 5A, 0 min). However, within 5 min of C2 removal, the HA signal altered dramatically to appear as distinct, punctate foci, indicating coalescence of the SERA6 into multiple intracellular condensates or aggregates (Fig 5A, 5 min). The RBCM had yet to rupture in these schizonts, as indicated by the continuous ankyrin signal, although it was unclear whether PVM rupture had occurred. Examination of ruptured schizonts from a later time point showed a barely detectable HA signal, suggesting that the foci eventually dispersed during egress (Fig 5A, 15 min). To gain more insights into the distribution of the foci, we used super-resolution fluorescence microscopy to acquire Z-stack images from individual schizonts. This confirmed that the foci were distributed throughout the schizont volume, although a few were co-localised with ankyrin, indicating association with the RBC cytoskeleton (Fig 5B). Schizont samples co-stained with antibodies to the major merozoite surface protein MSP1 did not indicate any specific associations between the foci and individual merozoites (Fig EV5A). To determine the composition of these foci with regard to the different SERA6 species formed during egress, we examined *SERA6-mTAP: loxP* schizonts arrested with E64-d and RAP-treated *SERA6-Cys644Ala-myc3 [SERA6-mTAP:loxP]* parasites, where SERA6 processing would largely be limited to production of p65-mTAP or p65-myc3, respectively. Distinct foci were observed in both cases (Fig EV5B and C), suggesting that the complexes likely contained the SERA6 p65 species.

The role of MSA180 as a protein cofactor of SERA6 prompted us to consider that MSA180 might also display dynamic localisation during egress. To explore this, *MSA180-HA3 [B11]* schizonts were sampled at different time points during egress (once again facilitated by the use of C2 washout) and fixed parasites examined by IFA. This revealed that immediately prior to SUB1 discharge into the PV, the MSA180-HA3 signal largely conformed to the "wagon wheel" pattern characteristic of soluble PV proteins, although distinct foci were also observed at the periphery of the HA signal (Fig 5C, 0 min), suggesting partial aggregation of the protein. These aggregates were likely still contained within the PV as there was no discernible overlap with the host RBC ankyrin signal. Following washout of C2, a range of modifications of the "wagon wheel" HA signal pattern was observed (Fig 5C), likely due to slight

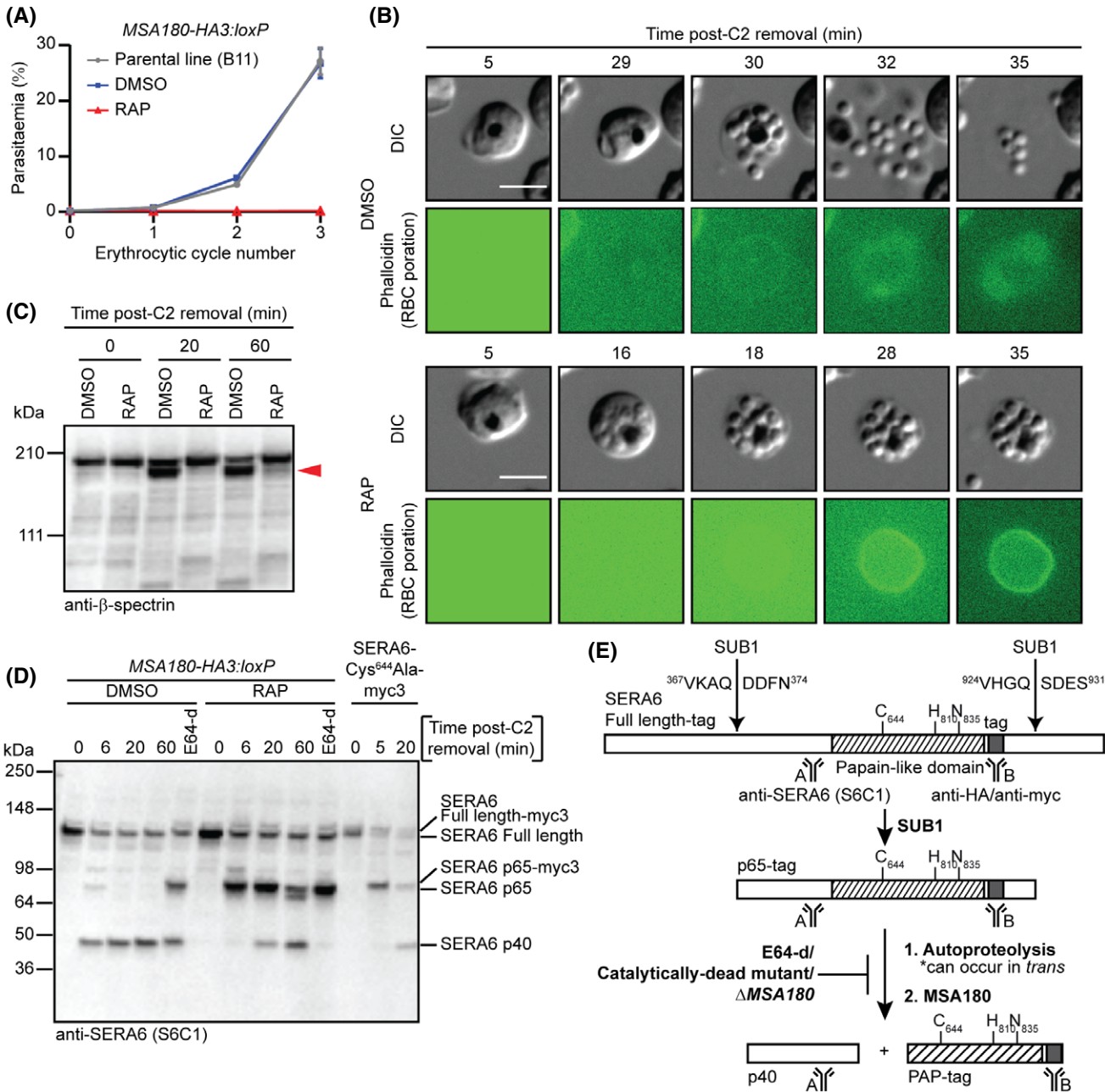

**Figure 4.  MSA180 is an essential cofactor required for maturation of SERA6, host RBC β-spectrin cleavage and egress.**

A   Replication of DMSO- and RAP-treated *MSA180-HA3:loxP* parasites over three erythrocytic cycles. Parasitaemia values are averages from replicates in different blood sources. The similar growth rate of DMSO-treated parasites and the parental B11 clone shows that the genetic modifications to generate *MSA180-HA3:loxP* parasites did not affect parasite viability. Error bars, ± SD (B11: *n* = 3; DMSO- and RAP-treated *MSA180-HA3:loxP*: *n* = 6 each).

B   Stills from simultaneous time-lapse DIC and fluorescence microscopic examination of DMSO- and RAP-treated *MSA180-HA3:loxP* cycle 0 schizonts following removal of C2-arrest (representative of 2 independent experiments). Note the defect in RBCM rupture and merozoite release in RAP-treated parasites despite normal PVM rupture (indicated by a sudden increased visibility and motility of merozoites) and RBCM poration (green ring indicating access and binding of phalloidin to the host RBC cytoskeleton). Scale bars, 5 μm.

C   Western blot of DMSO- and RAP-treated *MSA180-HA3:loxP* parasites showing that host RBC β-spectrin truncation (red arrowhead) does not occur in the absence of MSA180 (reproducible in 2 independent experiments). Schizonts were sampled at the indicated times following removal of C2-arrest.

D   Time-course comparing fate of SERA6 from DMSO- and RAP-treated *MSA180-HA3:loxP* parasites (representative of 2 independent experiments). Schizonts were sampled at the indicated times following removal of C2-arrest, or arrested with E64-d (50 μM). RAP-treated *SERA6-Cys644Ala-myc3 [SERA6-mTAP:loxP]* parasites were included in the blot to allow comparison with processing of catalytically dead SERA6 (SERA6-Cys644Ala-myc3). Note the accumulation of SERA6 p65 and poor conversion to SERA6 p40 in the RAP-treated (*ΔMSA180*) parasites.

E   Schematic model of SERA6 maturation.

Source data are available online for this figure.

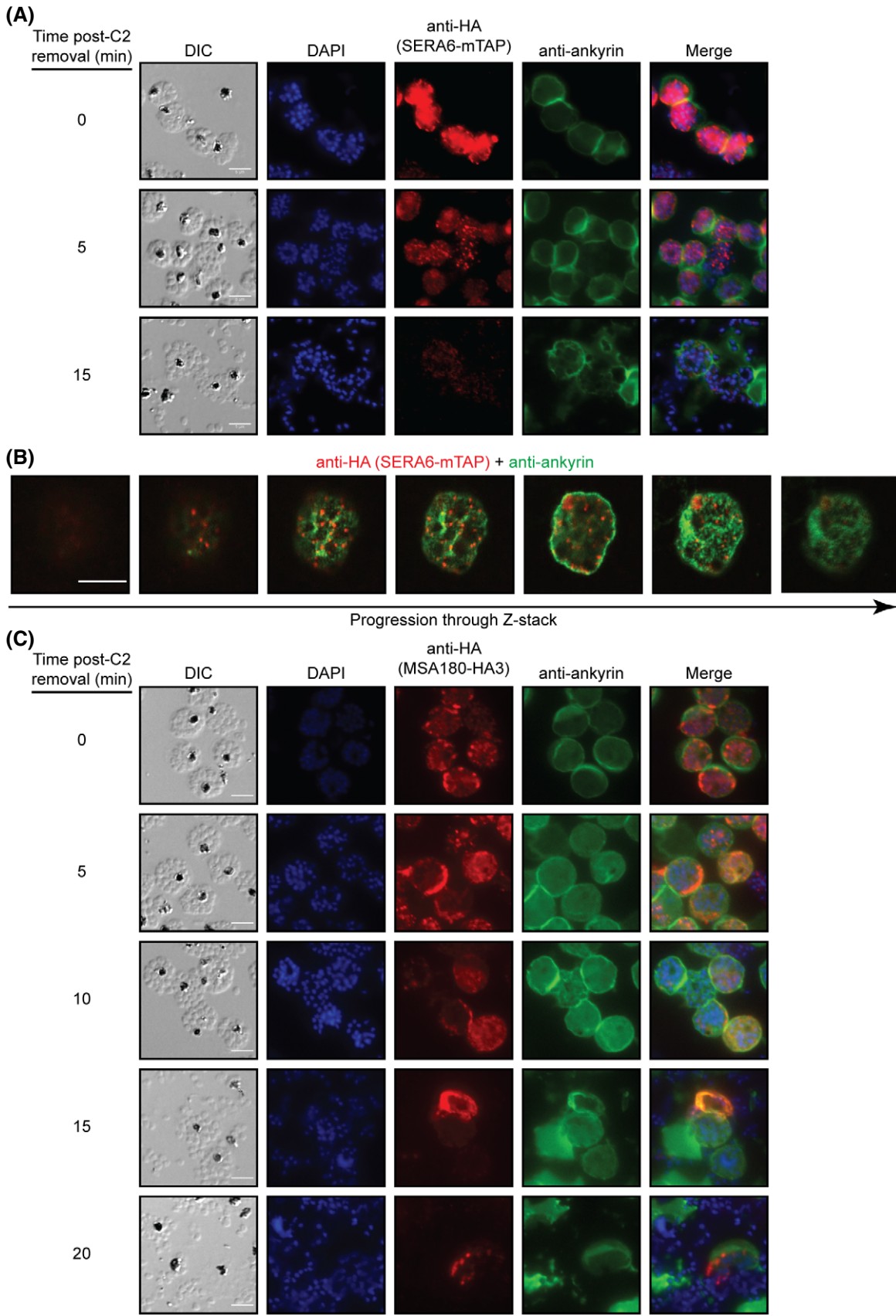

**Figure 5.**

**Figure 5.   SERA6 and its partner MSA180 form transient multimolecular aggregates during egress.**

A    Representative IFA images demonstrating the dynamic changes in localisation of SERA6-mTAP during egress (reproducible in 4 independent experiments). *SERA6-mTAP:loxP* schizonts were sampled at the indicated times following removal of C2-arrest, fixed and co-stained with DAPI (blue), anti-HA (red) and anti-ankyrin (green; ankyrin is a major component of the RBC cytoskeleton). Scale bar, 5 μm.

B    *Z*-stack series of a representative schizont demonstrating that SERA6 aggregates observed during egress are distributed throughout the schizont. *SERA6-mTAP:loxP* schizonts sampled 5 min following removal of C2-arrest were fixed and co-stained with anti-HA (red) and anti-ankyrin (green), then imaged on an Olympus FLUOVIEW FV3000. Scale bar, 5 μm. See also Fig EV5.

C    Representative IFA images demonstrating localisation of MSA180-HA3 during egress. *MSA180-HA3 [B11]* schizonts were sampled at the indicated times following removal of C2-arrest, fixed and co-stained with DAPI (blue), anti-HA (red) and anti-ankyrin (green). Scale bar, 5 μm.

asynchronicity of the schizonts and exacerbated by the analysis being performed on a collection of snapshots of different populations of schizonts. Of particular note was co-localisation of the HA signal with ankyrin, suggesting association of MSA180 with the RBC cytoskeleton, which persisted (though with a generally less intense signal over time) even in the residual vesiculated RBCM "ghosts" visible following parasite egress. Despite extensive IFA imaging of actively rupturing schizonts, we observed no indications of the anti-HA signal being associated with the surface of free merozoites, again supporting our conclusion that MSA180 is expressed as a soluble PV protein rather than on the merozoite surface. In the light of our co-precipitation data showing that mature SERA6 associates with two discrete fragments of MSA180 (including the region of MSA180-HA3 expected to possess the C-terminal epitope tag), we concluded that the SERA6-MSA180 complexes undergo transient aggregation into large-multimolecular complexes during egress, perhaps to facilitate SERA6 maturation and/or RBCM rupture.

**SERA6 maturation is inhibited by drug-like compounds**

Our identification of the peptidyl epoxysuccinate E64-d as an inhibitor of SERA6 maturation (Fig 2B and Appendix Fig S1) suggested that the egress-inhibitory properties of this compound might be mediated through direct inhibition of the autocatalytic step that generates the active PAP form of SERA6. E64-d has previously shown promise in clinical trials as an experimental treatment for muscular dystrophy (Miyahara *et al,* 1985a, 1985b; Watanabe *et al,* 1986), so does have drug potential. However, to seek less peptidic small molecules with similar activity, we searched the literature for descriptions of compounds displaying similar egress-inhibitory characteristics. This alerted us to a recent targeted screen of the

Medicines for Malaria Venture (MMV) Pathogen Box compound set which singled out the purine nitrile MMV676881 (Fig 6A) as an egress inhibitor that blocked RBCM rupture, but not PVM rupture or RBCM poration (Dans *et al,* 2020). This phenotype, which we confirmed here (Fig 6B), is remarkably similar to that of *ΔSERA6* and *ΔMSA180* parasites, suggesting that MMV676881 might act through inhibition of SERA6 maturation. To test this hypothesis, we monitored the effects of MMV676881 on SERA6 maturation and β-spectrin cleavage in *SERA6-mTAP:loxP* schizonts. As shown in Fig 6C and D, MMV676881 efficiently inhibited both proxies of SERA6 activity and was > 5-fold more potent than E64-d. These results strongly suggest that the egress-inhibitory activity of MMV676881 is through direct inhibition of SERA6 maturation.

To seek further structure-activity data, we developed a simple cell-based assay to screen a focused library of structurally related compounds, leading to the identification of a triazine nitrile (referred to as compound 31 in Mott *et al,* (2010)) (Fig 6A) as a similarly potent inhibitor of SERA6 maturation, β-spectrin cleavage and RBCM rupture (Fig 6B–D). Further examination of E64-d, MMV676881 and compound 31 showed that the compounds were also effective at blocking β-spectrin cleavage (Fig 6D) and egress (Fig 6E) in another human-infecting species, *P. knowlesi*, presumably by targeting the *P. knowlesi* orthologue of SERA6. Collectively, our results show that maturation of SERA6 can be efficiently targeted by drug-like molecules to prevent malaria parasite egress.

## Discussion

Our findings illustrate that maturation of SERA6 is a complex process involving a proteolytic cascade initiated by SUB1. This

**Figure 6.   SERA6 maturation is a druggable process.**

A    Structures of two compounds identified as inhibitors of SERA6 autoprocessing and egress.

B    E64-d, MMV676881 and compound 31 inhibit RBCM rupture and merozoite release despite normal PVM rupture, indicated by loss of a continuous ring of EXP2-mNeon signal (green) and RBCM poration (red ring indicating access and binding of phalloidin to the host RBC cytoskeleton) (reproducible in 2 independent experiments). *P. falciparum EXP2-mNeon* schizonts were allowed to reach the point of egress in the presence of C2 (1 μM), E64-d (10 μM), MMV676881 (2 μM), compound 31 (2 μM) or DMSO (0.1% v/v; control) then examined by DIC and fluorescence microscopy. Scale bar, 5 μm.

C    Western blot comparing dose response of E64-d, MMV676881 and compound 31-mediated inhibition of maturation of SERA6-mTAP from *SERA6-mTAP:loxP* parasites (reproducible in 3 independent experiments).

D    Western blot of *P. falciparum* and *P. knowlesi* parasites showing that host RBC β-spectrin truncation during egress (red arrowhead) is inhibited by all three *P. falciparum* SERA6-maturation inhibitors (concentrations as in panel B), matching the egress inhibition in panel E (reproducible in 2 independent experiments).

E    Giemsa-stained thin films showing inhibition of rupture of *P. knowlesi* schizonts in human RBCs by E64-d, MMV676881 or compound 31 (concentrations as in panel B; representative of 2 independent experiments). The increased space between individual merozoites still contained within the host RBC in parasites treated with E64-d, MMV676881 and compound 31 (compared to C2-arrested parasites) suggests that PVM rupture has occurred. Note that *P. knowlesi* schizonts generally form fewer merozoites (~10–12) than *P. falciparum* schizonts. Scale bar, 5 μm.

Source data are available online for this figure.

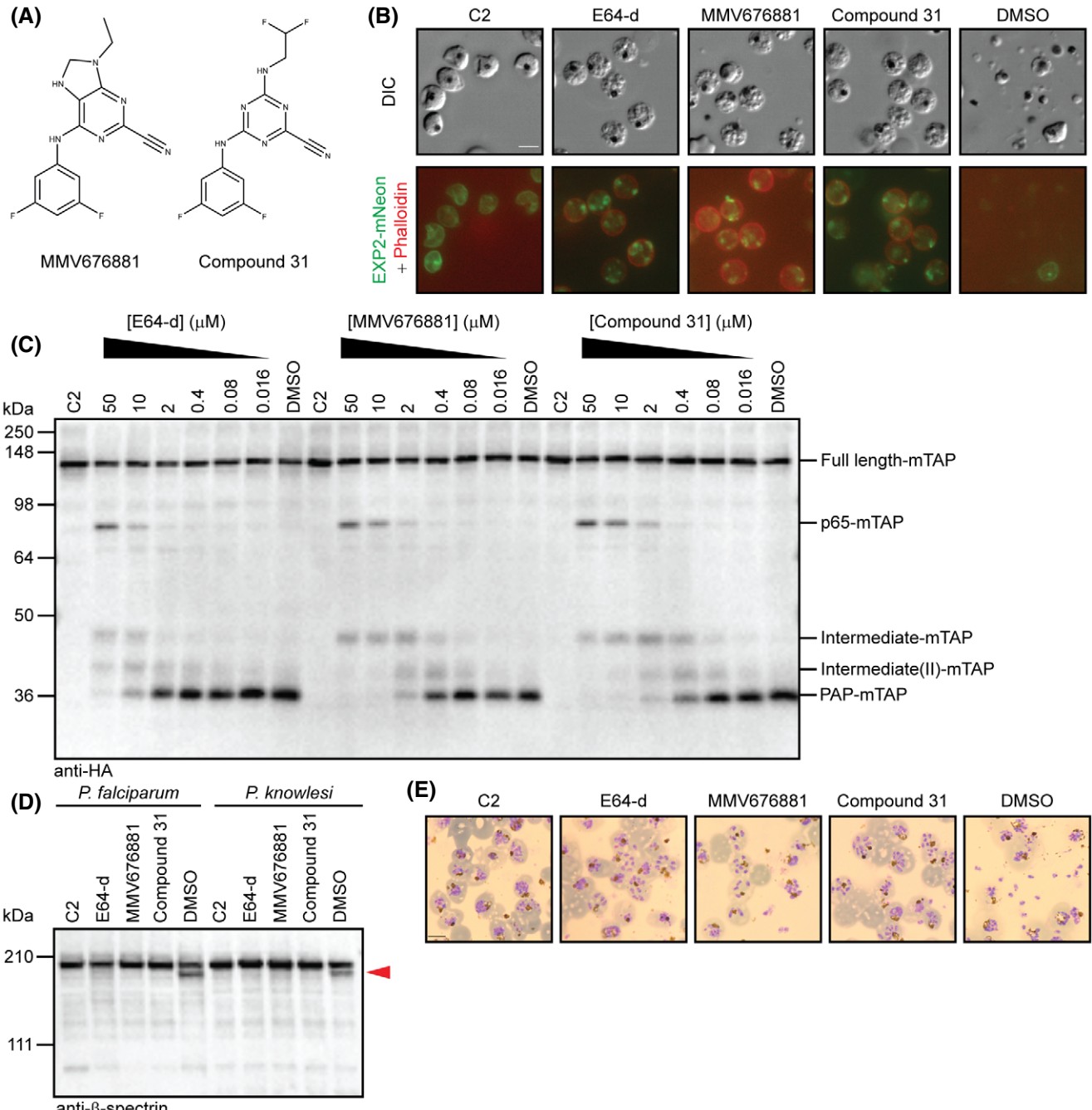

**Figure 6.**

triggers SERA6 autoproteolytic processing, which is facilitated by a previously unrecognised protein cofactor, MSA180, to ultimately produce a mature SERA6 species essentially comprised of its core catalytic domain. MSA180 is itself also processed by SUB1, indicating synchronised maturation within the PV of both SERA6 and its protein cofactor. We were able to confidently assign one SUB1 cleavage site within MSA180, but given the MSA180 processing profile, it is likely that it contains at least one additional SUB1 cleavage site. Our observations evoke a model in which maturation of SERA6 and its essential cofactor are temporally and functionally linked to

ultimately generate an active protease complex that mediates dismantling of the host RBC cytoskeleton (Fig 7). Egress is a rapid and highly choreographed process, so the multi-step nature of the SERA6 activation pathway may illustrate a need for tight regulation of the kinetics of the distinct molecular events that occur during egress to ultimately ensure effective dispersal of the released daughter merozoites. The importance of this for efficient parasite proliferation is exemplified by the phenotype of SERA5-null parasites, which display an egress defect characterised by premature membrane rupture but transient containment of daughter merozoites within

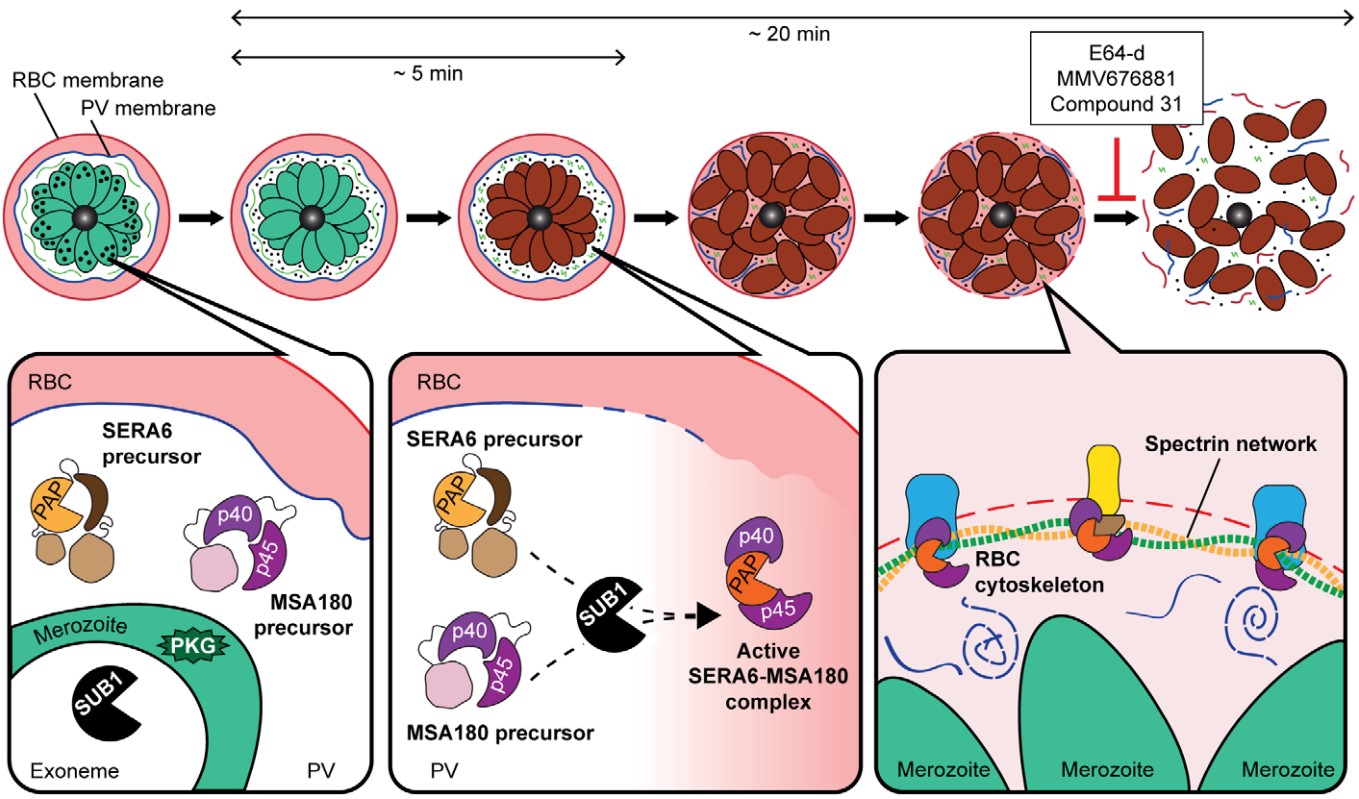

**Figure 7. Model of SERA6 maturation and function in merozoite egress.**

In mature schizonts, SERA6 and its essential protein cofactor, MSA180 are initially spatially separated from SUB1. The activation of the kinase PKG triggers discharge of SUB1 into the PV lumen where SUB1 initiates the coordinated maturation of both SERA6 and MSA180 by proteolysis, culminating in the formation of an active SERA6-MSA180 protease complex, perhaps in the form of multimeric aggregates. PVM rupture allows the active SERA6-MSA180 complex to access the RBC cytoskeleton, where it cleaves β-spectrin, detaching the spectrin network from the RBCM. This rapidly leads to RBCM rupture and merozoite egress.

the residual PVM and RBCM (Collins *et al*, 2017). This results in inefficient dissemination of the merozoites and severely reduces the parasite replication rate.

Why does SERA6 undergo a second, autocatalytic processing step following cleavage by SUB1? Recent X-ray crystal determination of the structurally related pseudoenzyme SERA5 (Smith *et al*, 2020) has revealed that the segment between the site 1 SUB1 cleavage site and the start of the papain-like domain in fact encompasses a prodomain. The globular bulk of this structure packs against the papain-like domain, whilst its C-terminal stalk extends across the "active site" cleft of SERA5 in a reverse orientation to a peptide substrate, similar to prodomains of other clan CA proteases such as cathepsin L (Turk *et al*, 2012). In the light of this, we can confidently postulate that the autocatalytic processing step that converts SERA6 p65 to PAP in fact corresponds to prodomain removal, and the released SERA6 p40 species likely represents the free prodomain. Since bound prodomain would be expected to impede access to substrate, this model is completely consistent with the terminal PAP species being the mature, proteolytically active form of SERA6.

The observed aggregation of SERA6 with its MSA180 partner during SERA6 maturation is intriguingly reminiscent of inflammasomes (Martinon *et al*, 2002) and apoptosomes (Srinivasula *et al*, 1998), large cytosolic multiprotein complexes that assemble rapidly to act as molecular platforms for recruitment and aggregation of procaspases and to facilitate their autoactivation into mature caspases. We speculate that MSA180 may function as a scaffold, perhaps seeding aggregation of SERA6 to create "egressomes" that provide localised foci of high enzyme concentration, promoting the rapid autoprocessing of SERA6 into its final PAP form in the short timescale between SUB1 discharge and RBCM rupture. The reversible nature of complex formation and disappearance could also occur via a phase separation process, shown in other eukaryotes to produce often transient membraneless organelles or biomolecular condensates that regulate the efficiency and rate of enzyme-catalysed reactions (Sehgal *et al*, 2020). Detailed characterisation of the molecular interactions between SERA6 and MSA180 will be required to conclusively elucidate the function(s) of MSA180, particularly as MSA180 does not contain recognisable protein domains. Such future work will be greatly facilitated by expression (or co-expression) of correctly folded recombinant forms of both proteins, an ambition that has to date proved elusive for SERA6.

In addition to being essential for SERA6 maturation, our pull-down data indicate that MSA180 remains associated with SERA6 following its conversion into the terminal PAP species (Fig 1B). The bound MSA180 fragments may be involved in maintaining SERA6 in a proteolytically active conformation, but it is also possible that MSA180 could play an additional role in substrate recognition by SERA6. Indeed, our observation that the SERA6-MSA180 aggregates

partly co-localise with the cytoskeletal protein ankyrin (Fig 5C) is particularly exciting as it aligns with the proposed role of SERA6 in cleaving key RBC cytoskeletal proteins (Thomas et al, 2018) to mediate RBCM rupture during egress. Whilst our discovery of an essential protein cofactor for SERA6 was unexpected, relationships of this nature are not unprecedented. As a well-studied example, the flavivirus NS3 serine protease requires a distinct viral protein, NS2B to act as a cofactor (Falgout et al, 1991; Zhang et al, 1992; Arias et al, 1993; Yusof et al, 2000). NS2B fulfils at least three distinct functions: (i) as a molecular chaperone to stabilise the NS3 protease (Erbel et al, 2006); (ii) to complete the substrate binding site of the resulting active protease complex (Erbel et al, 2006); and (iii) to localise the NS3 protease to its site of action (Clum et al, 1997). We hypothesise that MSA180 may play analogous roles.

In the process of advancing our understanding of the molecular mechanisms underlying the role of SERA6 in egress, we have developed a simple, robust cell-based approach to assay SERA6 activity by tracking its autoprocessing as well as cleavage of host RBC β-spectrin. Using these proxy readouts, we have identified compounds that will now serve as promising starting points to explore the potential of SERA6 as a drug target. Development of potent inhibitors of three other essential enzyme regulators of the egress pathway, PKG (Baker et al, 2017), SUB1 (Kher et al, 2014; Giovani et al, 2015) and an aspartic protease called plasmepsin X implicated in maturation of SUB1 (Nasamu et al, 2017; Pino et al, 2017; Zogota et al, 2019; Favuzza et al, 2020) are active research spheres. Combined inhibition of all four molecular effectors of egress could allow simultaneous targeting of consecutive, interdependent steps in the egress pathway, enhancing potency through additive or synergistic effects and reducing opportunities for selection of resistant parasites. SERA6-inhibitory compounds may also prove useful beyond the asexual blood stages of the malaria parasite life cycle. Other members of the SERA multigene family have been implicated in egress of liver-stage merozoites from hepatocytes to establish the erythrocytic cycle (Putrianti et al, 2020), as well as in parasite egress within the mosquito vector (Aly & Matuschewski, 2005), with implications for preventing disease transmission. These orthologues may be similarly susceptible to SERA6 inhibitors. Ultimately, pursuing SERA6 and egress as a drug target has potential to lead to an entirely new class of antimalarial drugs, expanding our repertoire of tools in the fight against malaria.

## Materials and Methods

### Reagents and antibodies

The antifolate drug WR99210 was from Jacobus Pharmaceuticals. Blasticidin-S (Sigma-Aldrich) was used as described previously (Collins et al, 2017). Rapamycin (RAP) (Sigma-Aldrich) was used at 50 nM. 4-[7-[(dimethylamino)methyl]-2-(4-fluorphenyl)imidazo[1,2-a]pyridine-3-yl]pyrimidin-2-amine (compound 2, C2) was kindly provided by Dr Simon Osborne (LifeArc) and stored at −20°C in DMSO as 10 mM stocks and used at a final concentration of 1 μM. N-(trans-epoxysuccinyl)-L-leucine 4-guanidinobutylamide (E64) and (2S,3S)-trans-Epoxysuccinyl-L-leucylamido-3-methylbutane ethyl ester (E64-d) (Sigma-Aldrich) were stored at −20°C in DMSO as 10 mM stocks and used at 50 μM final concentration unless stated otherwise.

Alexa Fluor 488 phalloidin and Alexa Fluor 594 phalloidin (Thermo Fisher Scientific) were used at 132 nM final concentration. The anti-HA mAb 3F10 (Roche) was used at a dilution of 1:1,000, as was the anti-myc mAb 4A6 (Merck). The β-spectrin-specific mAb VD4 (used at 1:1,000) and ankyrin-specific mAb H-4 (used at 1:100) were from Santa Cruz Biotechnology. Rabbit antisera to P. falciparum SERA6 (Ruecker et al, 2012), SERA5 (Collins et al, 2017) and AMA1 (Collins et al, 2009) have been described previously, as have the P. falciparum MSP1-specific mAbs 89.1 (Holder & Freeman, 1982) and X509 (Blackman et al, 1991). Rabbit polyclonal antisera against P. falciparum MSA180 were obtained commercially from Covalab, who performed peptide design, synthesis and rabbit immunisations. Briefly, two rabbits were immunised following an 88-day protocol involving four injections of synthetic peptides corresponding to Lys796-Ser812 (C-KDDHDLEH SAKNFYNIS-coNH2) and Glu1351-Tyr1366 (C-ESNSNAPQFNENYKEY-coNH2) supplemented with an equal volume of complete Freund's adjuvant.

### Parasite maintenance, synchronisation and transfection

The DiCre-expressing P. falciparum B11 line (Perrin et al, 2018) and P. falciparum EXP2-mNeon line (C. Bisson, manuscript in preparation) were maintained at 37°C in human RBCs in RPMI 1640 containing Albumax II (Thermo Fisher Scientific) supplemented with 2 mM ʟ-glutamine. The human RBC-adapted P. knowlesi A1.H1 line (Moon et al, 2013) was maintained as described previously (Mohring et al, 2019). Cultures were routinely microscopically examined using Giemsa-stained thin blood films and synchronised by standard procedures (Harris et al, 2005). Highly synchronous schizonts were isolated by centrifugation over 70% (v/v) isotonic Percoll (GE Healthcare, Life Sciences) cushions.

Transfections were performed by introducing DNA into ~$10^8$ Percoll-enriched schizonts by electroporation using an Amaxa 4D-Nucleofector X (Lonza), using program FP158 as previously described (Moon et al, 2013). For Cas9-based genetic modifications, 20 μg of Cas9 expression plasmid and 60 μg of linearised repair templates were electroporated. Drug selection with 2.5 nM WR99210 was applied 24 h post-transfection for 4 days. Clonal transgenic lines were obtained by serial limiting dilution in flat-bottomed 96-well plates (Thomas et al, 2016) followed by selection of single plaques. For sequential Cas9-based genetic modifications, established transgenic parasites were treated with 1 μM 5-fluorocytosine (5-FC) provided as clinical grade Ancotil® (MEDA) for 7 days prior to the secondary transfection. For transfections with plasmids to be maintained episomally, 10 μg of DNA was electroporated and the growth medium continuously supplemented with 2.5 μg/ml blasticidin-S from 24 h post-transfection. DiCre-mediated excision of DNA was induced by transient RAP treatment of highly synchronous early ring-stage parasites (2–3 h post-invasion) as previously described (Collins et al, 2013a). Parasite genomic DNA for genotype analysis was extracted using the QIAGEN DNeasy Bood and Tissue kit and analysed by PCR using GoTaq® G2 Green Master Mix (Promega).

### Plasmid construction

The endogenous SERA6 locus in the DiCre-expressing P. falciparum B11 clone was modified using Cas9-mediated genome editing to generate the SERA6-mTAP:loxP parasite line. To produce the repair

plasmid pJT_SERA6_repair, a DNA fragment containing in tandem: (i) a 5′ homology arm of 500 bp endogenous *SERA6* sequence; (ii) a recodonised *SERA6* sequence which encoded Cys720-Asn886 with a *loxPint* module (Jones *et al*, 2016) inserted between bp 2,934 and 2,935; (iii) a "mini" tandem affinity-purification (mTAP) tag (Stallmach *et al*, 2015) (inserted between codons Asn[886]-Val[887]); and (iv) a 3′ homology arm of 438-bp endogenous *SERA6* sequence was synthesised commercially (GeneArt, Thermo Fisher Scientific). The repair plasmid was linearized with NcoI overnight prior to transfection. To target the *SERA6* locus, three RNA targeting sequences (ACTATTTGGTAATTTAGGA, CACATACTATGGACACCTTT and ATAATATTAGCTGCATGGTC) were independently inserted into the pDC2 Cas9/gRNA/hDHFR (human dihydrofolate reductase)/yFCU (yeast cytosine deaminase/uridyl phosphoribosyl transferase)-containing plasmid (Knuepfer *et al*, 2017) (henceforth referred to as pDC2_Cas9) as described previously (Knuepfer *et al*, 2017) to generate three different Cas9 vectors.

Parasite line *MSA180-HA3:loxP* was created by sequential Cas9-mediated genome editing. The first modification, which generated line *MSA180-HA3 [B11],* was achieved using the repair construct pUC57_MSA180-HA3_rep. This contains in tandem: (i) a 5′ homology arm of 501-bp endogenous *MSA180* sequence; (ii) a recodonised *MSA180* sequence encoding Ile1231-Pro1455 immediately followed by a triple HA epitope sequence, a stop codon and a single *loxP* site; and (iii) a 3′ homology arm of 506 bp of the *MSA180* 3′UTR; the entire DNA fragment was obtained commercially (Genewiz). The repair plasmid was linearized with ScaI overnight prior to transfection. The required Cas9 vector was generated by inserting the RNA targeting sequence AAATAAAAATAAATATGCTA into pDC2_Cas9. Subsequent Cas9-mediated genome editing of *MSA180-HA3 [B11]* to generate the *MSA180-HA3:loxP* parasite line was achieved using the repair construct pUC57_MSA180_IntRep. The construct contains in tandem: (i) a 5′ homology arm of 500-bp endogenous *MSA180* sequence; (ii) a recodonised *MSA180* sequence encoding Lys234-Ala432 with a single *loxPint* module (Jones *et al*, 2016) inserted between bp 1,252 and 1,253; and (iii) a 3′ homology arm of 506-bp endogenous *MSA180* sequence; the entire DNA fragment was obtained commercially (Genewiz). The repair plasmid was linearized with XhoI overnight prior to transfection. The required Cas9 vector was generated by inserting the RNA targeting sequence AGAACATTATATATACGACG into pDC2_Cas9.

Construct pDC2_SERA6-WT-myc3 was generated by modifying the validated SERA6 complementation episome pDC2-wtSERA6 (Thomas *et al*, 2018) to insert a myc3 sequence between codons Asn[886]-Val[887]. Construct pDC2_SERA6-Cys644Ala-myc3 was generated by modifying pDC2-SERA6_Alamut (Cys644Ala) (Thomas *et al*, 2018) to insert a myc3 sequence between codons Asn[886] and Val[887].

## Egress time-course assay

Mature Percoll-enriched schizonts were incubated for ≥ 4 h at 37°C in complete medium supplemented with 1 μM C2, washed once in gassed protein-free medium pre-warmed to 37°C and supplemented with C2, then rapidly washed twice in similar medium either supplemented with 1 μM C2 or 50 μM E64/E64-d, or completely lacking either inhibitor, pelleting at 1,800 × *g* for 1 min. The cells were suspended at high density (~1 × 10⁹/ml) in the same medium as used for the final two washes and incubated at 37°C for the indicated durations. Schizont samples for Western blot analysis or immunoprecipitation were snap-frozen in liquid nitrogen and stored at −80°C until required whilst saponin-extracted fractions were obtained by resuspending pelleted schizonts in 4 volumes of 0.15% (w/v) saponin (Sigma-Aldrich) in protein-free medium. Following incubation at 37°C for 5 min, parasite-containing and soluble fractions were separated by centrifugation at 16,000 × *g* for 10 min at 4°C and both fractions were then snap-frozen in liquid nitrogen and stored at −80°C until required.

## Western blot and immunofluorescence

For Western blot analysis, frozen schizont suspensions or fractions were extracted directly into SDS and fractionated by SDS–PAGE then transferred onto nitrocellulose membranes, probed, visualised and documented as described previously (Thomas *et al*, 2018).

For IFA, air-dried thin blood films of *P. falciparum* parasite cultures were fixed in 4% (v/v) paraformaldehyde, permeabilised with 0.1% (v/v) Triton X-100 and probed with relevant primary antibodies as described previously (Thomas *et al*, 2018) then incubated with appropriate Alexa-conjugated secondary antibodies (Life Technologies). Slides were mounted using VECTASHIELD® Antifade Mounting Media with DAPI (Vector Laboratories). All images (unless stated otherwise) were acquired using a Nikon Eclipse Ni-E widefield upright microscope fitted with a Nikon N Plan Apo λ100×/1.45NA oil immersion objective (Nikon) and a Hamamatsu C11440 digital camera via the NIS Elements software (Nikon). Images were subsequently processed using Fiji (Schindelin *et al*, 2012).

## Parasite growth assays and parasitaemia quantitation by flow cytometry

Synchronous ring-stage parasites at 0.1% parasitaemia in 2% haematocrit were dispensed in triplicate into 12-well plates. 50 μl from each well was sampled at 0, 2, 4 and 6 d, stained with SYBR Green (Thermo Fisher Scientific) (diluted 1:10,000) and analysed by flow cytometry on a BD FACSVerse using BD FACSuite software. Data were analysed using FlowJo software. Samples were initially screened using forward and side scatter parameters and gated for RBCs. SYBR Green stain-positive RBCs were identified using a 527/32 detector configuration.

## Time-lapse DIC and fluorescence microscopy

Viewing chambers for live microscopy were constructed by adhering 22 × 64 mm borosilicate glass coverslips (VWR International) to microscope slides, as described previously (Collins *et al*, 2013a). Mature Percoll-enriched schizonts were incubated for ≥ 4 h at 37°C in complete medium supplemented with 1 μM C2. Subsequently, ~5 × 10⁷ schizonts were rapidly washed twice in 1 ml of gassed complete medium pre-warmed to 37°C and lacking C2, pelleting at 1,800 × *g* for 1 min. The cells were suspended in 60 μl of the same medium, either alone or supplemented with phalloidin and introduced into a pre-warmed viewing chamber which was then immediately placed on a temperature-controlled microscope stage held at 37°C on a Nikon Eclipse Ni-E wide-field microscope fitted with a Nikon N Plan Apo λ 100×/1.45NA oil immersion objective and a Hamamatsu C11440 digital camera and documented via the NIS

Elements software (Nikon). Images were acquired at 5- to 10-s intervals over a total of 30 min then processed and exported as TIFFs using FIJI (Schindelin *et al*, 2012).

### Immunoprecipitation and mass spectrometry analysis

Frozen schizont preparations were subjected to 3 freeze-thaw cycles in the presence of cOmplete™ Protease Inhibitor Cocktail (Roche) and 10 mM EDTA. Extracts were clarified by centrifugation at 16,000×g for 10 min at 4°C, filtered through 0.45 μm Costar® Spin-X® centrifuge tube filters (Corning), supplemented with 25 mM CHAPS (3-((3-cholamidopropyl) dimethylammonio)-1-propanesulfonate) (Thermo Fisher Scientific) and then incubated with Pierce™ Anti-HA Magnetic Beads (Thermo Fisher Scientific) at 4°C for 2 h. Sample processing followed the manufacturer's protocol. Protein complexes were eluted from beads by addition of reduced 1× SDS sample buffer. Samples were fractionated on 5–15% SDS–PAGE gels, stained with colloidal Coomassie and selected protein bands excised and washed. Reduced and alkylated proteins were in-gel digested with 100 ng trypsin (modified sequencing grade, Promega) or elastase overnight at 37°C. Supernatants were dried in a vacuum centrifuge and resuspended in 0.1% trifluoroacetic acid (TFA). 1–10 μl of digested protein acidified with 0.1% TFA was loaded at 15 μl/min onto a 2 × 0.3 mm Acclaim Pepmap C18 trap column (Thermo Scientific) on an Ultimate 3000 nanoRSLC HPLC (Thermo Scientific), prior to the trap being switched to elute at 0.25 μl/min through a 50 cm × 75 μm EasySpray C18 column. A 90-min gradient of 9−25% B over 37 min and then 25–40% B over 18 min was used followed by a short gradient to 100% B and back down to 9% B followed by equilibration in 9% B (buffer A: 2%ACN, 0.1% formic acid; buffer B: 80%ACN, 0.1% formic acid).

The Orbitrap was operated in "Data Dependent Acquisition" mode with a survey scan at a resolution of 120,000 from *m/z* 300–1,500, followed by MS/MS in "TopS" mode. Dynamic exclusion was used with a time window of 20 s. The Orbitrap charge capacity was set to a maximum of 1e6 ions in 10 ms, whilst the LTQ was set to 1e4 ions in 100 ms. Raw files were processed using Proteome Discoverer 2.1 (Thermo Fisher Scientific) against PlasmoDB (www.plasmodb.org) concatenated with the predicted SERA6-mTAP peptide sequence. A decoy database of reversed sequences was used to filter false positives, at a peptide false detection rate of 1%. Semi-tryptic peptides were validated by manual inspection of the MS/MS spectrum.

### Peptide cleavage assays

Synthetic peptide Ac-KVTGDSVENI-COOH was synthesised in-house by the Crick Peptide Chemistry Science Technology Platform, stored at −20°C in DMSO as a 40 mM stock and used at a final concentration of 400 μM. Peptide cleavage assays by rPfSUB1 and product identification by reversed phase HPLC and mass spectrometry were as described previously (Withers-Martinez *et al*, 2012).

### SERA6 inhibitor screen

A cell-based assay was developed to screen for egress-inhibitory compounds, based on spectrometric detection of the haemoglobin released from schizonts upon egress. Test compounds were prepared as 2 mM stocks in DMSO and diluted into gassed RPMI

1640 lacking phenol red and supplemented with 2 mM L-glutamine (Thermo Fisher Scientific) in round-bottomed 96-well plates. All test compounds were assayed at a final concentration of 10 μM. Wells containing C2 (1 μM), E64-d (10 μM) or DMSO (0.5% v/v) were included as controls. Mature Percoll-enriched *SERA6-mTAP:loxP* schizonts were incubated for ≥ 4 h at 37°C in complete medium supplemented with 1 μM C2 then washed once in warm, gassed protein-free colourless medium supplemented with C2. Cells were quantified using a Countess™ II Automated Cell Counter (Invitrogen) and diluted to $2 \times 10^9$/ml with the same warm, gassed protein-free colourless medium supplemented with C2 then dispensed in 5 μl aliquots into the prepared assay plates and incubated at 37°C for 1 h to allow egress. Cells were pelleted at 2,400 ×g for 3 min and 100 μl of each supernatant was transferred into flat-bottomed 96-well plates for measurement of absorbance at 415 nm using a SpectraMax M5e Multi-Mode Microplate Reader (Molecular Devices), blanking with equivalent samples to which no schizonts were added. To assess the effects of selected compounds on SERA6 maturation, remaining schizont suspensions in the round-bottomed plates were extracted directly into SDS and stored at −20°C prior to SDS–PAGE fractionation and Western blot analysis.

## Data availability

The mass spectrometry data from this publication have been deposited to the ProteomeXchange Consortium via the PRIDE partner repository (Perez-Riverol *et al*, 2019) and assigned the identifier PXD024084 (https://www.ebi.ac.uk/pride/archive/projects/PXD024084/).

**Expanded View** for this article is available online.

### Acknowledgements
The authors are indebted to the Medicines for Malaria Venture (MMV) for providing the Pathogen Box compounds and for additional provision of compound MMV676881. This research was funded in part by the Wellcome Trust (grant 106239/Z/14/A). For the purpose of Open Access, the author has applied a CC BY public copyright licence to any Author Accepted Manuscript version arising from this submission. This work was also supported by funding to MJB from the Francis Crick Institute (https://www.crick.ac.uk/), which receives its core funding from Cancer Research UK (FC001043; https://www.cancerresearchuk.org), the UK Medical Research Council (FC001043; https://www.mrc.ac.uk/) and the Wellcome Trust (FC001043; https://wellcome.ac.uk/). M.S.Y.T was in receipt of a Francis Crick PhD studentship, as well as funding from the Francis Crick Idea to Innovation (i2i) programme (grant P2019-0015). The work was also supported by Wellcome ISSF2 funding to the London School of Hygiene & Tropical Medicine and by the intramural research programme of the National Center for Advancing Translational Sciences (NCATS), Division of Pre-Clinical Innovation, USA.

### Author contributions
MSYT, KK, CW-M, SAH, JAT, FH and EK conducted the experiments; MS and MDH provided resources and expertise; APS and MJB supervised the work and obtained funding; MSYT and MJB designed the experiments and wrote the paper.

### Conflict of interest
The authors declare that they have no conflict of interest.

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
