## [Review Process File · The EMBO Journal]

Autocatalytic activation of a malarial egress protease is druggable and requires a protein cofactor

Michele Tan, Konstantinos Koussis, Chrislaine Withers-Martinez, Steven Howell, James Thomas, Fiona Hackett, Ellen Knuepfer, Min Shen, Matthew Hall, Ambrosius P. Snijders, and Michael Blackman

DOI: [10.15252/embj.2020107226](https://doi.org/10.15252/embj.2020107226)

Corresponding author: Michael Blackman (mike.blackman@crick.ac.uk)

Review Timeline:

Submission Date:	4th Nov 20
Editorial Decision:	4th Dec 20
Revision Received:	12th Feb 21
Editorial Decision:	4th Mar 21
Revision Received:	11th Mar 21
Accepted:	23rd Mar 21

Editor: Ieva Gailite

Transaction Report:

Thank you for submitting your manuscript for consideration by The EMBO Journal. We have now received three referee reports on your manuscript, which are included below for your information.

As you will see from the comments, all reviewers appreciate the work and the quality of the data and recommend publication of the manuscript after a minor revision. Given these positive evaluations from three experts of the field, I would like to invite you to address the comments from all reviewers in a revised version of the manuscript.

We have extended our 'scooping protection policy' beyond the usual 3-month revision timeline to cover the period required for a full revision to address the essential experimental issues. This means that competing manuscripts published during revision period will not negatively impact on our assessment of the conceptual advance presented by your study. Please contact me if you see a paper with related content published elsewhere to discuss the appropriate course of action. I would also be happy to discuss the revision in more detail via email or phone/videoconferencing.

Please feel free to contact me if you have any further questions regarding the revision. Thank you for the opportunity to consider your work for publication. I look forward to receiving the revised manuscript.

Referee #1:

This is a terrific body of work from a lab that is a world leader in high-quality research into mechanisms of *Plasmodium falciparum* parasite egress from infected red blood cells. This report includes several novel findings that make a substantial contribution to the field: 1) The discovery of a second autocatalytic processing of the cysteine protease SERA6 that mediates degradation of the host red blood cell membrane protein beta-spectrin; 2) Engagement of SERA6 with a novel cofactor termed MSA180 that localizes predominantly to the parasitophorous vacuole; 3) Evidence that MSA180-null parasites cannot egress; and 4) The identification of a chemical class of SERA6 inhibitors that block beta-spectrin cleavage and parasite egress. The study reports an outstanding combination of molecular and cell biology approaches, biochemistry, and early drug discovery. The article provides compelling evidence that targeted inhibition of the key mediators of parasite egress provides an appealing path towards the development of novel therapeutics. I think this study will be of substantial interest to the field and has a breadth of science that will be very appealing to the EMBO J readership.

I have only very minor comments that require correction:

Page 11: It would help to state the SUB1 recognition motif

Page 11: The authors should also mention that the data in Figure 3 also suggest partial localization of MSA180 to the merozoite surface.

Page 19: The authors should clarify that MMV676881 is >5-fold more potent than E64-d (the word "more" was missing).

Referee #2:

Tan et al. investigate the proteolytic processing of SERA6, an essential cysteine protease for the final stage of Plasmodium egress from infected RBCs. Earlier studies by this group demonstrated that SERA6-mediated cleavage of the host cytoskeletal protein β -spectrin leads to destabilization of the RBC cytoskeleton, thereby causing RBC membrane rupture. Before becoming active, SERA6 needs to be proteolytically processed by SUB1. In this present study, the authors show that in *P. falciparum*, SERA6 activation extends beyond the SUB1-mediated cleavage steps and involves autocatalytic cleavage. Interestingly, this processing is rather complex and is aided by a PV resident protein cofactor MSA180. Conditional knockout of MSA180 in late stage parasites not only blocks the autocatalytic cleavage steps of SERA6 processing, but also phenocopies the defects due to SERA6 deletion. Pulldown of activated SERA6 from parasites co-immunoprecipitates two processed fragments from MSA180. Based on these findings, the authors propose a model where SERA6 activity is not only linked to its autocatalytic processing but also to the functional complex formation between SERA6 and MSA180. The authors go on to identify drug-like compounds that inhibit SERA6 activity and subsequently parasite egress. This is of particular interest for future antimalarial drug development efforts.

This is a well-executed study that provides exciting findings that constitute a substantial advance for the field. The findings presented expand our current knowledge of the mechanisms of protease activation and since it is highly likely that it can be extrapolated to other intracellular pathogens, the manuscript should be of broad interest. That being said, some of the presented figures need to be more clearly explained. Therefore, I would recommend that the authors address the following minor comments:

1. Page 5, line 20: The authors suggest that the p40 fragment of SERA6 detected with the anti-SERA6 antibody (fig. 1A) spans the region between the SUB1 cleavage site 1 and the central papain-like domain. However, the predicted molecular weight for this fragment is not 40 KDa. By my calculation, it is approximately 27 kDa. Why does the fragment show aberrant migration in the western blot?
2. In fig. 1B, please state what the control is.
3. Fig S1B and 2B show complete excision of the SERA6 gene fragment and hence loss of SERA6 expression in parasites upon treatment with rapamycin. Given the fact that conversion of SERA6 p65 to the intermediate processed form is autocatalytic, please explain how the catalytic dead SERA6 still undergoes some processing (as shown in fig 2B, top panel, right)?
4. Page 11, line 10: The authors mention that SUB1-mediated cleavage of MSA180 generates two fragments, the central 40 KDa and the C-terminal 45 KDa. In fig. 3A, the authors mapped only one SUB1 cleavage site in MSA180. As shown, this cleavage could generate the 45 KDa fragment. Is there a predicted additional cleavage site to generate the 40 kDa fragment in MSA180?
5. For SUB1 processing of MSA180, it is too bad that a direct test was not done in the SUB1 cKO line, in this otherwise very comprehensive paper, though I would not insist on this experiment.
6. Despite the complete excision of part of the MSA180 gene upon rapamycin treatment, SERA6 still undergoes processing to the p40, indicating some level of autocatalytic activity is still retained. As the N-terminal end of MSA180 bears partial homology with the two downstream conserved regions (fig. 3A), could the residual SERA6 autocatalytic activity be maintained by complex formation between the N-terminal fragment of MSA180 (as this fragment would be retained in the parasites following rapamycin treatment) and SERA6 processed forms?
7. In figure 4D, eventually some p40 is generated in the RAP-treated culture. Have the authors followed by time-lapse out to 60 minutes post-C2 removal? If egress is completely blocked, perhaps this is more evidence that MSA180 is important for activity on spectrin, not just SERA processing.
8. In fig. 5C, even at 15 minutes post C2 wash, MSA180 shows strong colocalization with ankyrin. Given the proposed model that the SERA6-MSA180 complex interacts with the host cytoskeleton,

why did the authors observe no such colocalization between SERA6 and ankyrin at the same time point (fig. 5A)? Is it the images selected?

9. Labelling in fig. S1D is incorrect. The label says MSP1 but the legend says SERA5.

10. In fig. S6C, the green channel is labelled as SERA5, while the figure legend says it is anti-ankyrin.

11. In figure 1B, the authors used E64 to inhibit SERA6 activity and to arrest the RBC rupture, while for other figures they used the esterified form E64d. Is there any specific reason for switching to E64d?

12. There is no indication that any experiment was done more than once. In particular, 6c provides comparison of inhibitor potency, which cannot be judged from a single gel.

13. There is no metadata for any of the MS results.

14. In the second sentence of the discussion, "which requires" should probably be "which is facilitated by". Clearly MSA180 is essential, but it could be most important for SERA6 activity or for another function.

Referee #3:

Egress from infected red blood cells by the malaria parasite *Plasmodium falciparum* is a critical event that relies on a series of carefully coordinated steps including the secretion and activation of two essential proteases, namely SUB1 and SERA6. The current work builds on several previous important studies by the Blackman lab and others, and shows that SERA6 is initially processed by SUB1 but then undergoes autocatalytic activation in trans. This autocatalytic activation is required for egress, is aided by a protein cofactor (MSA180) identified in study and is blocked by two small molecule inhibitors that were previously not known to target SERA6. Overall, this work is thorough, carefully conducted, and significantly advances the field by yielding additional mechanistic details of the proteolytic cascade underlying parasite exit from erythrocytes. Also, the work likely corrects earlier work on the protein cofactor in terms of its localization and function. In this reviewer's view, the study has no major flaws and is recommended as a high priority for publication.

Minor comments:

1. One of the citations includes the first author's first name and middle initial: Christine R. Collins.

2. The findings suggest that MSA180 is required for SERA6 autoactivation, implying that there must be some sort of interaction between MSA180 and SERA6 prior to SERA6 autoactivation. The authors note co-immunoprecipitation of the MSA180 40kDa and 45kDa products in the 20 min post-release sample (Figure 1B). However, it appears as though the 45kDa species (I) might be co-IP'd in the E64-d blocked sample too. The authors should comment on whether they detected MSA180 in the E64-d blocked sample, and if so then some discussion of a potential interaction prior to autocatalytic activation of SERA6 seems warranted.

Referee #1:

This is a terrific body of work from a lab that is a world leader in high-quality research into mechanisms of Plasmodium falciparum parasite egress from infected red blood cells. This report includes several novel findings that make a substantial contribution to the field: 1) The discovery of a second autocatalytic processing of the cysteine protease SERA6 that mediates degradation of the host red blood cell membrane protein beta-spectrin; 2) Engagement of SERA6 with a novel cofactor termed MSA180 that localizes predominantly to the parasitophorous vacuole; 3) Evidence that MSA180-null parasites cannot egress; and 4) The identification of a chemical class of SERA6 inhibitors that block beta-spectrin cleavage and parasite egress. The study reports an outstanding combination of molecular and cell biology approaches, biochemistry, and early drug discovery. The article provides compelling evidence that targeted inhibition of the key mediators of parasite egress provides an appealing path towards the development of novel therapeutics. I think this study will be of substantial interest to the field and has a breadth of science that will be very appealing to the EMBO J readership.

I have only very minor comments that require correction:

Page 11: It would help to state the SUB1 recognition motif

This information has now been inserted into the revised manuscript, and we cite our previous work describing this (PMID: 21220481 and PMID: 22543039).

Page 11: The authors should also mention that the data in Figure 3 also suggest partial localization of MSA180 to the merozoite surface.

We agree with the reviewer that the IFA data alone do not rule out partial localisation with the merozoite surface. This section of the text has now been modified accordingly.

Page 19: The authors should clarify that MMV676881 is >5-fold more potent than E64-d (the word "more" was missing).

We apologise for this important omission, which has now been corrected.

Referee #2:

*Tan et al. investigate the proteolytic processing of SERA6, an essential cysteine protease for the final stage of Plasmodium egress from infected RBCs. Earlier studies by this group demonstrated that SERA6-mediated cleavage of the host cytoskeletal protein β -spectrin leads to destabilization of the RBC cytoskeleton, thereby causing RBC membrane rupture. Before becoming active, SERA6 needs to be proteolytically processed by SUB1. In this present study, the authors show that in *P. falciparum*, SERA6 activation extends beyond the SUB1-mediated cleavage steps and involves autocatalytic cleavage. Interestingly, this processing is rather complex and is aided by a PV resident protein cofactor MSA180. Conditional knockout of MSA180 in late stage parasites not only blocks the autocatalytic cleavage steps of SERA6 processing, but also phenocopies the defects due to SERA6 deletion. Pulldown of activated SERA6 from parasites co-immunoprecipitates two processed fragments from MSA180. Based on these findings, the authors propose a model where SERA6 activity is not only linked to its autocatalytic processing but also to the functional complex formation between SERA6 and MSA180. The*

authors go on to identify drug-like compounds that inhibit SERA6 activity and subsequently parasite egress. This is of particular interest for future antimalarial drug development efforts.

This is a well-executed study that provides exciting findings that constitute a substantial advance for the field. The findings presented expand our current knowledge of the mechanisms of protease activation and since it is highly likely that it can be extrapolated to other intracellular pathogens, the manuscript should be of broad interest. That being said, some of the presented figures need to be more clearly explained. Therefore, I would recommend that the authors address the following minor comments:

1. Page 5, line 20: The authors suggest that the p40 fragment of SERA6 detected with the anti-SERA6 antibody (fig. 1A) spans the region between the SUB1 cleavage site 1 and the central papain-like domain. However, the predicted molecular weight for this fragment is not 40 KDa. By my calculation, it is approximately 27 kDa. Why does the fragment show aberrant migration in the western blot?

The reviewer is correct that the p40 SERA6 fragment migrates much faster on SDS PAGE than predicted from its primary sequence. It is in fact fairly common for *Plasmodium* proteins to migrate aberrantly on SDS PAGE, especially when their primary sequence displays low complexity (which is especially frequent in *P. falciparum*). In this particular case, we suspect that the aberrant migration may be primarily due to the acidic nature of this species; the predicted pI of the SERA6 sequence between the SUB1 cleavage site 1 and the central papain-like domain (Asp371-Lys605) is ~4.4 (see https://web.expasy.org/compute_pi/). This is now briefly mentioned in the revised Fig 1A legend.

2. In fig. 1B, please state what the control is.

This control (similar pull-downs from parasites expressing un-tagged SERA6) is now described in the revised Fig. 1 legend.

3. Fig S1B and 2B show complete excision of the SERA6 gene fragment and hence loss of SERA6 expression in parasites upon treatment with rapamycin. Given the fact that conversion of SERA6 p65 to the intermediate processed form is autocatalytic, please explain how the catalytic dead SERA6 still undergoes some processing (as shown in fig 2B, top panel, right)?

The reviewer correctly notes that, upon RAP-treatment of the SERA6-Cys644Ala-myc3 [SERA6-mTAP:loxP] parasites conversion of the episomally-expressed, catalytically dead SERA6-Cys644-Ala-myc3 protein to the 'Intermediate-myc3' form is only slightly affected. However, we see no conversion to the PAP form, which we believe to be the catalytically active mature form of SERA6. We interpret this result as indicating that conversion to the Intermediate form is not autocatalytic. This notion is supported by our observation that even at the highest concentrations used, the small molecule inhibitors described in Fig. 6 do not prevent formation of the intermediate form. We speculate that formation of the Intermediate form may be an artefact of alternative cleavage, possibly due to slow, low-level cleavage at (an) additional site(s) by SUB1.

We did not explicitly draw attention to this point in the original manuscript, but in response to the reviewer's comment this is now mentioned in the revised paragraph describing Fig 2B. The key conclusion here - which is not altered by this observation - is that no beta-spectrin cleavage or egress occurs under condition where formation of PAP is prevented, consistent with our central model that autocatalytic maturation of SERA6 to PAP is required for protease activity, RBC cytoskeletal breakdown and egress.

4. Page 11, line 10: The authors mention that SUB1-mediated cleavage of MSA180 generates two fragments, the central 40 kDa and the C-terminal 45 kDa. In fig. 3A, the authors mapped only one SUB1 cleavage site in MSA180. As shown, this cleavage could generate the 45 kDa fragment. Is there a predicted additional cleavage site to generate the 40 kDa fragment in MSA180?

The reviewer raises a very pertinent point. We have obviously searched for additional potential SUB1 cleavage site(s) within the MSA180 sequence. This is not completely straightforward, since although known SUB1 sites show a clear consensus, there is a degree of redundancy at several positions. We are currently addressing this question experimentally by generating recombinant MSA180; we anticipate that treatment of this protein with recombinant PfSUB1 will reveal additional cleavage sites. For now, in the present manuscript, we would prefer not to speculate on this. However, in response to the reviewer's question we have now modified the revised Discussion (first paragraph of this section) to mention that we consider it likely that (an)other SUB1 site(s) exist within MSA180.

5. For SUB1 processing of MSA180, it is too bad that a direct test was not done in the SUB1 cKO line, in this otherwise very comprehensive paper, though I would not insist on this experiment.

The reviewer suggests an interesting and potentially very informative experiment. Whilst we have obviously considered this, the experiment would require epitope-tagging MSA180 on the genetic background of our SUB1 conditional knockout line, so would not be completely trivial. We certainly will explore this in the future, but are grateful that the reviewer does not insist on including this experiment in the present manuscript.

6. Despite the complete excision of part of the MSA180 gene upon rapamycin treatment, SERA6 still undergoes processing to the p40, indicating some level of autocatalytic activity is still retained. As the N-terminal end of MSA180 bears partial homology with the two downstream conserved regions (fig. 3A), could the residual SERA6 autocatalytic activity be maintained by complex formation between the N-terminal fragment of MSA180 (as this fragment would be retained in the parasites following rapamycin treatment) and SERA6 processed forms?

We thank the reviewer for this thoughtful suggestion. We had not considered this possibility, but it seems plausible. However, we have decided not to further complicate an already complex manuscript by laying out this hypothesis in the revised paper.

7. In figure 4D, eventually some p40 is generated in the RAP-treated culture. Have the authors followed by time-lapse out to 60 minutes post-C2 removal? If egress is completely blocked, perhaps this is more evidence that MSA180 is important for activity on spectrin, not just SERA processing.

Although we did not follow the parasites by time-lapse out to 60 min post-C2 washout, we did examine beta-spectrin cleavage at that time point (Fig 4C) and there was no cleavage. We therefore assume that PAP was not produced under these conditions (although we were not able to examine this directly since SERA6 was not tagged in the MSA180-HA3:loxP line).

We agree that that this could indicate that MSA180 may be important not just for SERA6 maturation but also for the proteolytic activity of SERA6 on beta-spectrin, and our speculative hypothesis (mentioned in the 4th paragraph of the Discussion) that MSA180 could play a role in substrate recognition by SERA6 takes this notion into account.

8. In fig. 5C, even at 15 minutes post C2 wash, MSA180 shows strong colocalization with ankyrin. Given the proposed model that the SERA6-MSA180 complex interacts with the host cytoskeleton, why did the authors observe no such colocalization between SERA6 and ankyrin at the same time point (fig. 5A)? Is it the images selected?

As always with IFA, the images throughout this figure were selected to show the dominant pattern observed at each time point. By 15 min post-C2 washout, most of the SERA6 and MSA180 signal had disappeared from the vast majority of the schizonts (this can be seen by examining the other schizonts in the image); schizonts displaying residual strong MSA180 labelling at 15 min were rare occurrences, and we suspect that this is what the reviewer is referring to here. We have now amended the relevant part of the text to make it clear that the schizont-associated SERA6 and MSA180 signals generally became less intense over time following egress.

9. Labelling in fig. S1D is incorrect. The label says MSP1 but the legend says SERA5.

We apologise for this error, which has now been corrected.

10. In fig. S6C, the green channel is labelled as SERA5, while the figure legend says it is anti-ankyrin.

We apologise for this error, which has now been corrected.

11. In figure 1B, the authors used E64 to inhibit SERA6 activity and to arrest the RBC rupture, while for other figures they used the esterified form E64d. Is there any specific reason for switching to E64d?

We apologise for the omission of this detail. The switch to use of E64-d was instigated by our observation that it is a more potent inhibitor of SERA6 maturation than E64, probably as a result of its better membrane-permeability. The revised manuscript now mentions this in the Fig 2 legend, and the relevant supporting data are provided in a new Appendix Fig S1.

12. There is no indication that any experiment was done more than once. In particular, 6c provides comparison of inhibitor potency, which cannot be judged from a single gel.

We apologise for this omission. The revised manuscript now provides figures for numbers of replicates for each experiment.

13. There is no metadata for any of the MS results.

The mass spectrometry data have now been uploaded to an appropriate repository; full details are now provided in the revised manuscript.

14. In the second sentence of the discussion, "which requires" should probably be "which is facilitated by". Clearly MSA180 is essential, but it could be most important for SERA6 activity or for another function.

We agree with this. This change has now been made in the revised manuscript.

Referee #3:

Egress from infected red blood cells by the malaria parasite Plasmodium falciparum is a critical event that relies on a series of carefully coordinated steps including the secretion and activation of two essential proteases, namely SUB1 and SERA6. The current work builds on several previous important studies by the Blackman lab and others, and shows that SERA6 is initially processed by SUB1 but then undergoes autocatalytic activation in trans. This autocatalytic activation is required for egress, is aided by a protein cofactor (MSA180) identified in study and is blocked by two small molecule inhibitors that were previously not known to target SERA6. Overall, this work is thorough, carefully conducted, and significantly advances the field by yielding additional mechanistic details of the proteolytic cascade underlying parasite exit from erythrocytes. Also, the work likely corrects earlier work on the protein cofactor in terms of its localization and function. In this reviewer's view, the study has no major flaws and is recommended as a high priority for publication.

Minor comments:

1. One of the citations includes the first author's first name and middle initial: Christine R. Collins.

This has now been corrected.

2. The findings suggest that MSA180 is required for SERA6 autoactivation, implying that there must be some sort of interaction between MSA180 and SERA6 prior to SERA6 autoactivation. The authors note co-immunoprecipitation of the MSA180 40kDa and 45kDa products in the 20 min post-release sample (Figure 1B). However, it appears as though the 45kDa species (I) might be co-IP'd in the E64-d blocked sample too. The authors should comment on whether they detected MSA180 in the E64-d blocked sample, and if so then some discussion of a potential interaction prior to autocatalytic activation of SERA6 seems warranted.

Our evidence clearly suggests that there is no physical interaction between full-length MSA180 and full-length SERA6 prior to cleavage of the two proteins by SUB1. This is based on our observation that MSA180 is not pulled down with SERA6 from parasites in which the entire egress pathway has been arrested with C2 (i.e. prior to SUB1 discharge into the PV; see the '0 min' lane of Fig 1B). In the E64-blocked samples, where SUB1-mediated conversion of SERA6 to p65 occurs but there is no further autocatalytic conversion to PAP, we agree that low levels of the MSA180 fragments appeared to be co-IPed on the Coomassie-stained gels (Fig 1B, 'E64' lane)). However, in 3 separate pull-down experiments we were unable to reproducibly confirm that suspicion by either mass spectrometry or immunoblotting. So whilst we agree in principle with the reviewer's intuition, we believe that - on the basis of the available evidence - we have to exercise caution in suggesting that the MSA180 p45/p40 complex can form with SERA6 p65. Whilst mechanistically important and of interest, our uncertainty about this detail does not alter our major conclusions in the present manuscript that (1) conversion of SERA6 to PAP involves an autocatalytic step and (2) that MSA180 is required for SERA6 maturation and function. In the revised manuscript we have therefore not altered our final model.

Thank you for submitting a revised version of your manuscript. Your revised study has now been seen by one of the original referees, who finds that their main concerns have been addressed and recommend publication of the manuscript. Therefore, I would like to invite you to address the remaining editorial issues before I can extend the official acceptance of the manuscript.

Referee #2:

The reviewers' comments have been addressed by the authors and the revised manuscript is suitable for publication. The work is well done and exciting.

The authors performed the requested editorial changes.

Editor accepted the revised manuscript.

YOU MUST COMPLETE ALL CELLS WITH A PINK BACKGROUND ↓
PLEASE NOTE THAT THIS CHECKLIST WILL BE PUBLISHED ALONGSIDE YOUR PAPER

Corresponding Author Name: Michael J Blackman

Journal Submitted to: The EMBO Journal

Manuscript Number: EMBOJ-2020-107226